# THE POWER OF LLM-GENERATED SYNTHETIC DATA FOR STANCE DETECTION IN ONLINE POLITICAL DISCUSSIONS

**Stefan Sylvius Wagner**[1,3]**, Maike Behrendt**[1,3]**, Marc Ziegele**[1] **& Stefan Harmeling**[2,3]
[1]Heinrich Heine University Düsseldorf, [2]Technical University Dortmund
[3]Lamarr Institute for Machine Learning and Artificial Intelligence
{stefan.wagner, maike.behrendt, marc.ziegele}@hhu.de
stefan.harmeling@tu-dortmund.de

## ABSTRACT

Stance detection holds great potential to improve online political discussions through its deployment in discussion platforms for purposes such as content moderation, topic summarization or to facilitate more balanced discussions. Typically, transformer-based models are employed directly for stance detection, requiring vast amounts of data. However, the wide variety of debate topics in online political discussions makes data collection particularly challenging. LLMs have revived stance detection, but their online deployment in online political discussions faces challenges like inconsistent outputs, biases, and vulnerability to adversarial attacks. We show how LLM-generated synthetic data can improve stance detection for online political discussions by using reliable traditional stance detection models for online deployment, while leveraging the text generation capabilities of LLMs for synthetic data generation in a secure offline environment. To achieve this, (i) we generate synthetic data for specific debate questions by prompting a Mistral-7B model and show that fine-tuning with the generated synthetic data can substantially improve the performance of stance detection, while remaining interpretable and aligned with real world data. (ii) Using the synthetic data as a reference, we can improve performance even further by identifying the most informative samples in an unlabelled dataset, i.e., those samples which the stance detection model is most uncertain about and can benefit from the most. By fine-tuning with both synthetic data and the most informative samples, we surpass the performance of the baseline model that is fine-tuned on all true labels, while labelling considerably less data.

## 1 INTRODUCTION

With the recent advent of powerful generative Large Language Models (LLMs) such as ChatGPT, Llama (Touvron et al., 2023) and Mistral (Jiang et al., 2023), new ways of performing stance detection have opened up via zero-shot or chain-of-thought prompting. This is especially important in the area of online political discussion where topics are complex and labelled data is hard to come by. At the same time, an ever important use case in online political discussions is being able to use stance detection for an ongoing discussion to, e.g., suggest suitable comments for engagement between participants (Küçük & Can, 2020; Behrendt et al., 2024). In the case of LLMs, while strong at analysing complex topics and at open-ended text generation, explicit classification can be inconsistent (Cruickshank & Xian Ng, 2023), they are prone to biases (Ziems et al., 2023) and open to adversarial attacks (Greshake et al., 2023). More traditional stance detection models based on, e.g., BERT (Devlin et al., 2019) are task-specific and therefore consistent in their output, however they need large amounts of labelled data (Mehrafarin et al., 2022; Vamvas & Sennrich, 2020) to perform well.

In this work, we combine both traditional stance dectection and LLMs to get the best of both worlds. For stance dectection, we use BERT as a lightweight stance detection model that produces fast and consistent output given the data it has been fine-tuned on. To address the issue of needing large amounts of data, we propose to generate synthetic data with an LLM to augment the stance detection model for fine-tuning. This allows us to leverage LLMs in an offline setting to enhance classical

Figure 1: **We investigate the use of LLM-generated synthetic data for stance detection in online political discussions.** (A) We generate synthetic data ●○ for specific questions using a Mistral-7B model. The synthetic data is then used to fine-tune the stance detection model. We show that fine-tuning with synthetic data improves the performance of the model, since the synthetic data is roughly faithful to the real data's ◆◇ underlying distribution. However, some real world samples ◇ cannot be captured by the synthetic data. (B) We therefore use the synthetic data to identify the most informative samples ◆ in the unlabelled real data pool, which are better off labelled by human experts. Combining the synthetic data with the manually labelled most informative samples improves the performance of the model even further.

stance detection models, which are better suited and safer for use in an online setting. Furthermore, we show that the synthetic data allows us to gain insights about the real world data. We show that the synthetic data generated by the LLM can serve as a reference distribution for stance detection, since the LLM is able to generate high quality samples which fall distinctly into the respective stance classes. This has two benefits: (i) we can analyse the data for potential biases by comparing the alignment of the synthetic data distribution to the real world data distribution. (ii) we can tackle the issue of extracting ambiguous samples that are difficult for the model to classify and therefore deteriorate its performance. Since the synthetic data partitions the space between both classes, ambiguous samples that the stance detection model cannot classify properly can be identified as lying in between the two classes. These samples can then be labelled manually. Fine-tuning with these additional samples, outperforms the model that is fine-tuned on all true labels alone, while only having labelled a subset of samples in the unlabelled data pool. We illustrate our method in Figure 1.

We view stance detection as a binary classification problem (*favor* or against), where we explore the following questions:

**(Q1) Does fine-tuning a stance detection model with synthetic data improve stance detection performance?** We first analyse whether fine-tuning the BERT model with synthetic data improves stance detection and show that this approach almost reaches the model trained with all true labels and is superior to using zero-shot Mistral-7B for the complex topics in online political discussions. *We reveal that a stance detection model can be tailored to a certain topic with only synthetic data.*

**(Q2) How does the synthetic data improve performance and does it align with real world data?** Our second question analyses the generated synthetic data. We analyse how well it aligns with the real world training data by visualising the T-SNE projected embeddings of the stance detection model and by comparing the entropy distribution of the synthetic data to the real data. *We find that the synthetic data aligns well with the real data, indicating that the LLM is able to generate comments for both stances while introducing minimal further bias.*

**(Q3) Can we further improve the model by using the synthetic data as labelled reference distribution for active learning?** The synthetic data allows us to identify unlabelled real data samples that improve the model even further through active learning. Due to the canonical nature of the synthetic data, we are able to extract real word samples for human labelling that are difficult (ambiguous) for the model to classify. We do this, by determining the $k$-nearest synthetic neighbours of the real data. *The stance detection model is fine-tuned jointly with these samples and the synthetic data, where we surpass the baseline model even when it is fine-tuned on all true labels, while labelling considerably less data manually.*

## 2   BACKGROUND

**Stance Detection for Online Political Discussions.**   Stance detection, a sub-task of sentiment analysis (Romberg & Escher, 2023) and opinion mining (ALDayel & Magdy, 2021), aims to automatically identify an author's stance (*favor*, *against*, or *neutral*) towards a discussed issue or target. In online political discussions, this involves determining if the contribution in question is *for* or *against* a topic like tax increases. Stance detection has been identified as an important task for improving discussion summarization (Chowanda et al., 2017), detecting misinformation (Hardalov et al., 2022), and evaluating opinion distributions in online political discussion and participation processes (Romberg & Escher, 2023). Stance detection is also used in recommender systems and discussion platforms (Küçük & Can, 2020). Still, due to its dependency on context, stance detection is a highly challenging task. Identifying stance requires understanding both the question and the contributor's position, complicated by users often deviating from the original question and discussing multiple topics in the same thread (Ziegele et al., 2014), leading to little usable training data. Some works in stance detection use graph convolutional networks to learn more out of the present data (Zhang et al., 2022; Li & Goldwasser, 2019). Recently, fine-tuning transformer-based models (Vaswani et al., 2017; Liu et al., 2022) to solve stance detection is a common practice, but training these models requires a large amount of annotated data, which for the large variety of questions in online political discussions is unfeasible to acquire.

**Active Learning.**  The aim of *active learning* is to minimize the effort of labelling data, while simultaneously maximizing the model's performance. This is achieved by selecting a *query strategy* that chooses the most interesting samples from a set of unlabelled data points, which we refer to as *most informative* samples. These samples are then passed to, e.g., a human annotator for labelling. There exist many different query strategies such as Query By Comittee (QBC, (Seung et al., 1992)), Minimum Expected Entropy (MEE, Holub et al. (2008) or Contrastive Active Learning (CAL, Margatina et al. (2021)). By actively choosing samples and asking for the correct labelling, the model is able to learn from few labelled data points, which is advantageous especially when annotated datasets are not available. Within the domain of political text analysis, many different tasks lack large amounts of annotated data. It has been already shown in the past that these tasks can benefit from the active learning: e.g., stance detection (Kucher et al., 2017), topic modeling (Romberg & Escher, 2022), speech act classification (Schmidt et al., 2023) or toxic comment classification (Miller et al., 2020). In this work, we examine how LLM-generated synthetic data can be used instead of real labelled data to select the most informative samples to be manually labelled.

**Stance detection and synthetic data generation with LLMs.** Recent work has shown that synthetic data generated from LLMs can be used to improve the performance of a model on downstream tasks. Møller et al. (2023) showed that synthetic data can be used to improve the performance of a model on downstream classification tasks by comparing the performance of a model finetuned on LLM-generated data to crowd annotated data. In many cases the model finetuned on LLM-generated data outperforms the model finetuned on crowd annotated data. Mahmoudi et al. (2024) study the use of synthetic data for data augmentation in stance-detection. The authors use GPT-3 to generate synthetic data for a specific topic with mixed results due to the inability of GPT-3 to generate good data samples. In our work,we use a newer LLM model, Mistral-7B, which generates better synthetic data samples and show that we can generate synthetic data that matches the real data distribution. Veselovsky et al. (2023) analyse in which ways synthetic data is best generated for tasks like sarcasm detection and sentiment analysis. The authors reach the conclusion that grounding the prompts to generate the synthetic data to real samples helps improve the quality of the synthetic data. Similarly, Li et al. (2023) argue that subjectivity in the classification task determines whether synthetic data can be used effectively. It has been shown that LLMs can be used directly for stance detection such as (Cruickshank & Xian Ng, 2023), (Burnham, 2023) (Ziems et al., 2023). However, the general conclusion of these studies is that while LLMs are competitive with other transformer models such as BERT, especially for edge cases, they exhibit replication issues. (Burnham, 2023) also discuss the posibility of pre-training models on more specific data to improve the generalisation capability of the model. Ziems et al. (2023) highlight the potential biases that can emerged in open ended generation tasks and classification performance varies depending on how representative the training data is. We therefore focus on using LLMs to generate synthetic data to solve key challenges in stance detection such as the lack of available data for specific topics and labelling large amounts of data, rather than using LLMs directly for the task.

## 3 METHOD

In line with our declared contributions, we present our core ideas to improve the performance of the stance detection model: (i) To fine-tune the model with synthetic data, we first define the synthetic dataset and show our pipeline to generate it. The baseline model is then further fine-tuned on the synthetic data. (ii) We then present our synthetic data-based approach to identify the most informative samples from a batch of unlabelled data, where we propose a synthetic extension to the QBC (Query by Commitee, (Seung et al., 1992)) method, where the synthetic data act as an ensemble of experts. As described in Section 1 and Figure 1, we use the synthetic data as reference distribution to identify the most informative samples in the unlabelled data pool. The idea is that for ambigouous samples the k-synthetic nearerst neighbours are split in their labels and therefore lie on the decision boundary of the model.

### 3.1 PRELIMINARIES

Political discussions are typically centered around questions $q \in \mathcal{Q}$ (sometimes also called issues or targets). For stance detection, we usually have for each of these questions $q$ a set of labelled data $\mathcal{D}^{(q)} = \{(x^{(i)}, y^{(i)})\}_{i=1}^{I}$ where $x^{(i)} \in \mathcal{X}$ is a statement (or comment) and $y^{(i)}$ is the stance of the statement, with $y^{(i)} \in \{0, 1\} = \mathcal{Y}$. Note, that we use the notation $\mathcal{D}^{(q)}$ for labelled and for unlabelled datasets (then the labels are ignored). We view the stance detection model as a binary classification function $f : \mathcal{Q} \times \mathcal{X} \to \mathcal{Y}$, where we included the question as input to provide context. The stance detection model such as BERT (Devlin et al., 2019) is *fine-tuned* by minimizing the cross-entropy loss between the predicted labels $\hat{y}^{(i)} = f(q, x^{(i)})$ and the actual labels $y^{(i)}$.

### 3.2 GENERATING SYNTHETIC DATA FOR STANCE DETECTION

To generate synthetic samples, we employ a quantized version of the Mistral-7B-instruct-v0.1 model to generate comments on a specific question $q$, using the following prompt:

```
A user in a discussion forum is debating other users about the following question:
[q] The person is in favor about the topic in question.  What would the person
write?  Write from the person's first person perspective.
```

where "[q]" must be replaced with the question $q$. Similarly, to generate a negative sample, we replace "is in favor" with "is not in favor". As in the X-Stance dataset (Vamvas & Sennrich, 2020), we assign the two labels 0 and 1. We denote the question-specific synthetic dataset as:

$$\mathcal{D}_{\text{synth}}^{(q)} = \left\{ (x_{\text{synth}}^{(m)}, 1) \right\}_{m=1}^{M/2} \cup \left\{ (x_{\text{synth}}^{(m)}, 0) \right\}_{m=1+M/2}^{M} \tag{1}$$

where half of the $M$ synthetic data samples have *positive* labels, i.e., are comments in *favor* for the posed question, while the other half is *against*. Since the dataset is in German, we translate the questions $q$ with a "NLLB-300M" (NLLB Team et al., 2022) translation model. The English answers from the Mistral-7B model are then translated back to German using the translation model. We also tried other similar sized open source LLMs (Llama, Openassistant, Falcon), but found that only the Mistral-7B model produced sensible comments.

Overall, the generated dataset $\mathcal{D}_{\text{synth}}^{(q)}$ will be used in two ways: (i) to augment the existing dataset $\mathcal{D}^{(q)}$ in order to increase the amount of training data, and (ii) to detect the most informative samples in the unlabelled data pool, which is explained next.

### 3.3 GETTING THE MOST INFORMATIVE SAMPLES: SYNTHETIC QUERY BY COMITTEE

To identify the ambiguous (most informative) samples as described in **(Q3)** we take from two active learning methods: Query by Comitee (QBC) (Seung et al., 1992) and Contrastive Active Learning (CAL, Margatina et al. (2021)). Instead of using QBC's ensemble of experts and the KL-divergence based information score in CAL, we directly use the synthetic data and its labels to identify ambigous samples using $k$ nearest neighbors. The most informative samples are then the data points with the most indecisive scores. Synthetic Query by Comitte (SQBC) consists of three steps:

**(1) Generate the embeddings.** Given some embedding function $g : \mathcal{Q} \times \mathcal{X} \to \mathcal{E}$, we generate embeddings for the unlabelled data, $E = \left\{ e^{(i)} \right\}_{i=1}^{I} = \left\{ g(q, x^{(i)}) \right\}_{i=1}^{I}$ and for the labelled synthetic

data $E_{\text{synth}} = \big\{ e_{\text{synth}}^{(m)} \big\}_{m=1}^{M} = \big\{ g(q, x_{\text{synth}}^{(m)}) \big\}_{m=1}^{M}$. Note that $q$ is the question for which we generate the synthetic data and for which we want to detect the most informative samples. If obvious from the context, we often omit the superscript $(q)$.

**(2) Using the synthetic nearest neighbours as oracles to score the unlabelled data.** For the $i$-th unlabelled embedding $e^{(i)}$ let $\text{NN}(i)$ be the set of indices of the $k$ nearest neighbours (wrt. to the embeddings using the cosine similarity) among the labelled embeddings $E_{\text{synth}}$. The score for each unlabelled data point counts the number of labels $y_{\text{synth}}^{(m)} = 1$ among the nearest neighbours, i.e.,

$$s(i) = \sum_{m \in \text{NN}(i)} y_{\text{synth}}^{(m)} \in \{0, \dots, k\}. \tag{2}$$

For our experiments, we choose $k = M/2$ which worked well across all experiments (other values for $k$ are possible, but did not lead to significantly better results).

**(3) Choosing the most informative samples.** The scores take values between 0 and $k$. For 0, the synthetic nearest neighbours all have labels $y_{\text{synth}}^{(m)} = 0$, for value $k$, all have labels $y_{\text{synth}}^{(m)} = 1$. The *most informative* samples have a score around $k/2$. We thus adjust the range of the scores so that values in the middle range have the smallest scores (close to 0). We do this by subtracting $k/2$ from the score and taking the absolute value,

$$s'(i) = |s(i) - k/2|. \tag{3}$$

The $J$ most informative samples $\mathcal{D}_{\text{MInf}}^{(q)} \subset \mathcal{D}^{(q)}$ among the unlabelled samples are the $J$ samples with the smallest scores. In the experiments we vary $J$ to study the impact of manually labelled most informative samples. Finally, the most informative samples are labelled by a human expert.

## 4 EXPERIMENTS

### 4.1 DATASETS

**X-Stance dataset.** We evaluate on the German dataset of the X-Stance dataset (Vamvas & Sennrich, 2020), which contains $48,600$ annotated comments on many policy-related questions (140 topics), answered by Swiss election candidates. We chose the German X-Stance dataset because it to our knowledge the most comprehensive stance detection dataset with a variety of topics and comments, while also being focused on online political discussions in governmental participation processes. Known english datasets such as SemEval-2016 (Mohammad et al., 2017) or P-Stance (Küçük & Can, 2020) offer less variety and are more focused on social media discussions. The comments are labelled either as being in *favor* (positive) or *against* (negative) the posed question. The dataset is split in training and testing questions, i.e,. a question in the training dataset does not appear in the test dataset. Furthermore, for each question $q$ from the training data, there are several annotated comments, which form the dataset $\mathcal{D}_{\text{train}}^{(q)}$. Analogously, for the test data we have a set of annotated comments written as $\mathcal{D}_{\text{test}}^{(q)}$. To refer to the whole training dataset we write $\mathcal{D}_{\text{train}} = \cup_{q \in \mathcal{Q}} \mathcal{D}_{\text{train}}^{(q)}$.

For our experiments, we fine-tune all stance detection models for each question separately allowing for better performance since the data distributions can vary greatly between questions (also a common scenario in online political discussions). To limit computation time, we selected 10 questions from the test dataset to evaluate our method, that best reflect the variability of the data (see Appendix E.1). We split the datasets of these questions into Test-Train and Test-Test to fine-tune the BERT model on synthetic data and to perform active learning. That is, we use the Test-Train dataset to fine-tune the model and the Test-Test dataset to evaluate the model. The number of comments in the 10 selected test questions is shown in Table 10.

**Synthetic dataset.** For synthetic data-augmentation and active learning based on SQBC (see Section 3.3) we generate synthetic datasets of varying sizes $M = \{200, 500, 1000\}$ for each of the 10 questions. The synthetic data follows the same structure as the data from the X-stance dataset, where for a specific question $q$ we have $M$ comments and $M$ labels. Each set contains $M/2$ positive labels and $M/2$ negative labels, i.e., the synthetic data is balanced. We show samples of the synthetic data in Appendix C.2.

|            | Q1        | Q2      | Q3       | Q4       | Q5       | Q6        | Q7       | Q8        | Q9       | Q10       |
|------------|-----------|---------|----------|----------|----------|-----------|----------|-----------|----------|-----------|
| Total-Test | 233 \| 267 | 29 \| 77 | 55 \| 141 | 102 \| 79 | 181 \| 88 | 216 \| 181 | 281 \| 97 | 178 \| 166 | 49 \| 130 | 169 \| 259 |
| Test-Train | 146 \| 154 | 19 \| 44 | 34 \| 83 | 68 \| 40 | 111 \| 50 | 130 \| 108 | 163 \| 63 | 100 \| 106 | 29 \| 78 | 98 \| 158 |
| Test-Test  | 87 \| 113 | 10 \| 33 | 21 \| 58 | 34 \| 39 | 70 \| 38 | 86 \| 73 | 118 \| 34 | 78 \| 60 | 20 \| 52 | 71 \| 101 |

Table 1: **Number of comments in our 10 selected test questions for our experiments.** The numbers are split into (favor | against) labels. We have a $60 - 40$ (Test-Train, Test-Test) split.

## 4.2 EXPERIMENTAL SETUP

**General setup.** For all experiments, we start with a pre-trained BERT base model and adapt to the stance detection task by fine-tuning on the X-Stance training dataset $\mathcal{D}_{\text{train}}$ (all questions). We call this the **Baseline** since it is the vanilla BERT-based stance detection (e.g., Vamvas & Sennrich (2020)).

We evaluate our methods along the lines of the questions proposed in Section 1: **(Q1)**: we analyse the effect of fine-tuning **Baseline** with synthetic data and compare it to the **Baseline** that was only fine-tuned on $D_{\text{train}}$. **(Q3)**: we fine-tune **Baseline** with synthetic data and the most informative samples. We present the baselines and our methods in the following:

**Baseline methods.**

- Baseline: the default BERT model fine-tuned only on $\mathcal{D}_{\text{train}}$, (e.g., Vamvas & Sennrich (2020)).
- True Labels: we fine-tune **Baseline** on the true labels of $\mathcal{D}_{\text{test}}^{(q)}$.
- Random+Synth, CAL+Synth: we use the active learning approaches to get the most informative samples $\mathcal{D}_{\text{MInf}}^{(q)}$.

**Our methods.**

- Baseline+Synth: we fine-tune the **Baseline** on the synthetic data $\mathcal{D}_{\text{synth}}^{(q)}$.
- True Labels+Synth: we fine-tune **True Labels** additionally on the synthetic data $\mathcal{D}_{\text{synth}}^{(q)}$.
- SQBC+Synth: we apply our active learning approach to get the most informative samples $\mathcal{D}_{\text{MInf}}^{(q)}$.

For fine-tuning with synthetic data only **(Q1)**, we compare the performance of our approaches to the baselines without synthetic data. For the active learning methods **(Q3)**, we compare to the non-active learning and active learning baselines. For further experimental details we refer to Appendix D.

**Analysing the synthetic data.** To analyse the synthetic data **(Q2)**, we visualize the BERT embeddings of the synthetic data together with the embeddings of the real world data (see Figure 3 and Appendix A). We use T-SNE (van der Maaten & Hinton, 2008) to project the embeddings, i.e., the output CLS token of the BERT model to a two-dimensional space. To assess whether the synthetic data captures the overall characteristics of the real world data and shares similar labels, we plot the individual embeddings of the synthetic data together with the means of the embeddings of the real world data. Additionally, we plot their corresponding labels. Finally, we also visualize the most informative samples selected by the different active learning methods.

## 4.3 RESULTS: EVALUATING THE EFFECTIVENES OF THE SYNTHETIC DATA

**(Q1) LLM-generated synthetic data substantially improves stance detection.** We show in Figure 2, that fine-tuning with only synthetic data improves the stance detection model. For $M = 1000$ the performance almost reaches the **True Labels** model, indicating that we can tailor a model to a certain topic without having any data for it. Furthermore, we can still improve the model consistently when new labelled data is available as seen by the **True Labels+Synth** models. In Section 5.1, we also analyse stance detection with zero-shot and fine-tuning approaches on Mistral-7B, but show that these are far less effective on the X-Stance dataset than our fine-tuned BERT model. In the following, we attempt to provide an understanding as to why the synthetic data is so effective.

**(Q2) The synthetic data aligns well with real world data.** We compare the T-SNE projected embeddings of the synthetic and real data in Figure 3(A) (more visualisations in Appendix A). The synthetic data aligns well with the real world dataset, since the means of the training data are close and in the direction of the synthetic data. We further analyse the synthetic data in Section 5.2: the

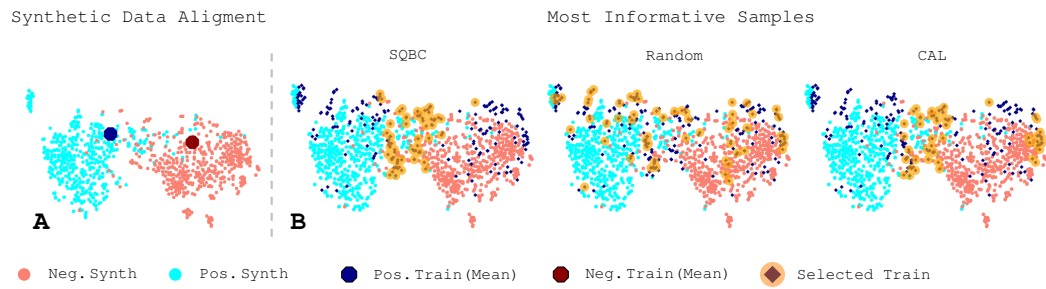

Figure 2: **Q1: Fine-tuning the model with synthetic data improves performance for increasing dataset size:** Shown are the F1-Score of fine-tuning with **Only Synthetic Data** (left) and **Synthetic Data + True Labels** (right) for increasing synthetic dataset size. Even if a dataset has been fully labelled, augmenting it with synthetic data proves equally as effective.

Figure 3: **(Q2) Analysing the synthetic data (M=1000):** The synthetic data aligns well with the real data, which is crucial for improving stance detection performance and to check for potential biases introduced by the synthetic data. SQBC selects the samples that are in between the two classes, i.e, that are the most ambiguous and informative for the model.

synthetic data is generally of high quality which validates the notion that it can serve as a good reference distribution for the model. From a statistical learning point of view the synthetic data can be thought of smoothening the decision boundary of the model. This also explains why fine-tuning with real samples is also very effective. Furthermore, we show that it is crucial to generate synthetic data that aligns with the given topic The insight that the synthetic data provides a good prior, serves as motivation to use the synthetic data to identify ambiguous samples that are the most informative to the model. We elaborate on this in the following section.

**(Q3) Synthetic data aids in finding unlabelled samples that further improve the stance detection model by extending its decision boundary.** We show the results of combining the most informative samples and synthetic data in Figure 4. Combining both, we ouperform **True Labels** while using *only* 25% of the labelled data. We compare the selection strategy of the methods in Figure 3(B): Due to the k-nearest neighbours objective of **SQBC**, the model selects samples that are in between the two classes, which proves superior to **CAL** and to **Random** for smaller synthetic data sizes. **CAL** performs the worst across the board: it assumes that similar embeddings that have different outputs are ambiguous, which makes it prone to outliers in the real data, e.g., when the stance detection model misclassifies a sample. Therefore, **CAL** often selects samples from only one class which worsens performance. Interestingly for $M = 1000$, **Random** outperforms both active learning methods **SQBC** and **CAL**. **Random** selects similar samples to **SQBC**, but also uniformly samples from outliers from both classes, extending the decision boundary of the model. We argue this is especially effective for larger synthetic dataset sizes where the synthetic data smoothens the decision boundary and thus mitigates the high variance introduced by the most informative samples. Thus, the model

Fine-tuning with most informative samples and synthetic data

Figure 4: **(Q3) Fine-tuning with synthetic data improves stance detection, while combining most informative samples and synthetic data surpasses the baseline model fine-tuned with all true labels (▬▬▬ above dashed line ‑‑‑) using less manually labelled data:** The reason for the performance increase can be attributed to two phenomena: (i) the synthetic data smoothens the decision boundary of the model making it more robust to outliers. (ii) The most informative samples improve the model where the synthetic data distribution is not expressive enough.

remain robust while extending the decision boundary. However, as we show in Section 5.2, the real data is quite homogenous. Therefore, with severe outliers present, **Random** could select these and worsen performance. This would not happen with **SQBC** due to its k-nearest neighbour objective.

## 5 Ablations

We investigate different aspects of using synthetic data for online political discussions. First, we study how well LLMs perform on classifying stance on the X-Stance dataset directly. Our assumption behind this is that while LLMs are strong at generating open ended text, they seem to have more difficulties when conditioned on a specific task combined with a narrow dataset. We also study the properties of the synthetic data by calculating the per comment entropy distribution toghether with the comment length. We then compare to the real data. Furthermore, to determine whether fine-tuning on a per topic basis is sensible, we study if the observed performance improvement is related to the content or the structure of the synthetic dataset.

### 5.1 Using LLMs directly for X-Stance

Cruickshank & Xian Ng (2023) and Gül et al. (2024) have shown promising results using zero-shot stance detection and fine-tuning various LLMs on common stance detection datasets such as SemEval-2016 (Mohammad et al., 2017) and P-Stance (Li et al., 2021). However, both these datasets have relatively little variety in topics, with the topics addressing more popular discussion topics such as general US-Politics. For this reason, we evaluate Mistral-7B on $D_{train}$ of X-Stance, which has a larger number of niche topics. We adopt the prompt and fine-tuning scenario (fine-tuning over 4 epochs with LoRA (Hu et al., 2021)) as in Gül et al. (2024) and use our Mistral-7B model for both zero-shot stance detection and fine-tuned stance detection.

Table 2 shows that zero-shot stance detection barely reaches the performance of the pre-trained BERT baseline. Suprisingly, fine-tuning the Mistral-7B model with $\mathcal{D}_{train}$ worsened performance even further. We tried various hyperparameter settings and fine-tuned for up to 10 epochs, more than the 4 used in Gül et al. (2024). We believe the poor performance can be attributed to a few reasons: (i) The topics in the X-Stance dataset are likely not present in the training sets of the Mistral 7-B model compared to SemEval and P-Stance which contain social media comments. (ii) Due to the smaller parameter count, the model may struggle to capture both the topic and the given comment for stance prediction. This often showed with the Mistral-7B model not giving consistent classification outputs or it would often refuse to predict stance. (iii) Furthermore, the model is not tailored to give

single line response. In fact, the responses were often verbose so we also accepted answers that contained the words "favor" or "against". Better prompting strategies could improve performance, however with our findings we believe that as of now using LLMs for open-ended text generation is more effective than conditioning them to give a specific output for stance detection. We also tried other similar sized open source LLMs (Llama, Openassistant, Falcon) and found that they struggled similarly in producing consistent zero-shot classification.

| F1 Score (avg. 10 Questions) | |
|---|---|
| Fine-tuned LLM | 0.182 |
| Zero-shot LLM | 0.419 |
| Baseline | 0.693 |
| Baseline+Synth (M=1000) | 0.723 |
| SQBC+Synth (M=1000) | 0.754 |

Table 2: **LLM-based stance detection vs BERT-based stance detection:** We compare the Mistral-7B performance to the our BERT stance detection models. We see that zero-shot stance detection barely reaches the pre-trained baselines' performance. Fine-tuning the LLM also proved difficult where the performance of the fine-tuned model worsened. Our findings for X-Stance suggest that LLMs are good at producing open-ended text, while struggling when being prompted to give a specific stance.

| F1 Score (avg. 10 Questions) | | | |
|---|---|---|---|
| M | 200 | 500 | 1000 |
| Baseline | 0.693 | 0.693 | 0.693 |
| Baseline+Synth | 0.711 | 0.717 | 0.723 |
| Baseline+Synth (Misaligned) | 0.699 | 0.704 | 0.694 |

Table 3: **Topic alignment is crucial for the synthetic data to be effective:** To determine whether improvement with synthetic data is due to the structure or content of the synthetic data, we augment the stance detection model with misaligned synthetic data. That is, the synthetic data does not align with the question given to the stance detection model. We observe the model only performs meaningfully better, when the synthetic data aligns with the posed question.

## 5.2 STUDYING THE SYNTHETIC DATA

**Properties of the synthetic data.** To determine the quality and diversity of the synthetic data, we calculate the entropy of each comment (entropy over words) for both the synthetic and real data and compare the interquartile ranges of the corresponding entropy distributions in Figure 5. We also determine the average comment length for the interquartile ranges. The entropy reveals information about the content of both datasets, while the length acts a surrogate for structure. We observe, that the entropies between the real and synthetic data are similar, while the average comment length of the synthetic data is longer than the real data's. Looking at the samples in Appendix C.1, we observe that the real comments have a concise (even emotional) writing style, which is common in online political discussions. The synthetic data comments are verbose and more reserved. Thus the difference in entropy could be attributed to the synthetic data containing less "emotional" comments compared to the real data. Nonetheless, we feel the synthetic samples manage to capture the content of the real data. Interestingly, the difference in length does not seem to affect the synthetic data alignment as can be seen from Figure 3. It is clear that the synthetic data is well aligned with the real data, even though the comment length between both sets of data differs. That is, the BERT model is largely invariant to the comment length and is more sensitive to the content of the comments rather than their structure.

**Content vs structure: Is per topic fine-tuning necessary?** To further validate our approach of fine-tuning the model per topic, we determine whether the improved performance comes from the discussion oriented structure of the data or from the content of the data. We fine-tune the model with misaligned questions and synthetic datasets, since we believe data across different topics does not necessarily share the same representation, justifying the need for per topic fine-tuning. Table 3 shows that fine-tuning with the synthetic data is only effective when the synthetic dataset is aligned with the posed question as can be seen by **Baseline+Synth**. Fine-tuning with synthetic data from a different topic **Baseline+Synth (Misaligned)** provides little improvement, delivering performance close to the **Baseline**. This validates the usefulness of the generated synthetic data for the stance detection model.

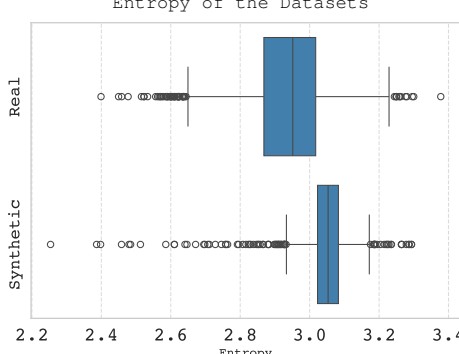

| Statistical Summary of Synthetic Data | | |
|---|---|---|
| Total samples | | 10000 |
| Number of outliers | | 261  (2% − 3%) |

| Entropy of Synthetic vs Real Data | | |
|---|---|---|
| Distr.  Range (Synthetic \| Real) | Avg.  Entropy | Avg.  Length |
| Minimum | 2.25 \| 2.39 | 3 \| 8 |
| 0% − 25% | 2.96 \| 2.78 | 101 \| 12 |
| 25% − 50% | 3.04 \| 2.91 | 124 \| 19 |
| 50% − 75% | 3.06 \| 2.98 | 134 \| 26 |
| 75% − 100% | 3.11 \| 3.07 | 142 \| 35 |
| Maximum | 3.27 \| 3.37 | 282 \| 53 |

Figure 5: **Synthetic data properties: (Left)** The per comment entropy of the synthetic data is similar to the entropy of the real data, where the synthetic data has a higher mean entropy than the real data, while the real data has higher variance. **(Right)** The generated synthetic data is of high quality, with only a few outliers. Interestingly, synthetic comments are longer than real ones, making real comments denser and synthetic ones more verbose. We argue that since the projected embeddings in Figure 3 show alignment between both datasets, BERT appears to be invariant to comment length.

## 6    DISCUSSION

**Potential impact of this work.** An apparent advantage of our approach is the possiblity of divding the synthetic data generation and the stance model fine-tuning. The former can be outsourced to dedicated infrastructure, while with the latter fine-tuning and inference is accessible even for smaller organisations with fewer resources. Considering the large amount of topics in participation processes, the ability to generate synthetic data or to reduce labelling effort with SQBC further increases the benefits for smaller organisations that can't afford large scale data collection or labelling efforts.

**Limitations.** One limitation of our approach is that we fine-tune a separate model for each question. While this leads to good results, a common approach is to fine-tune a single (and thus more general) model for several questions (like pre-training **Baseline**). However, visualising the synthetic data in Figure 3 and Appendix A, we observe that the underlying data distribution differs (sometimes greatly) for each question, which strongly suggests that each question benefits from fine-tuning a different model. This also aligns well with the per topic setting of online (political) discussions, considering that lightweight stance detection models can be fine-tuned in less than a minute even with a synthetic dataset size of $M = 1000$ on a reasonable GPU (NVIDIA A100). Another concern are biases that could be potentially introduced through the synthetic data. We addressed this in Section 4 by comparing the distributions of the synthetic data and real world data. We argue that analysing potential biases that could be introduced to the stance detection model through the synthetic data is easier in a single-question setting. In a multiple-question setting data from other topics could introduce biases into the model that are harder to detect. While effective for online discussions, due to the above observations, synthetic data quality should be assessed separately for different use cases.

**Future work** Despite this work focusing on smaller interpretable models, we believe future work should investigate why the Mistral-7B model performs poorly on the X-Stance dataset. We shared our thoughts around this in Section 5.2, but this requires more extensive study. Another avenue for future work could be about leveraging the synthetic data distribution to generate more novel synthetic data. For instance, with synthetic data as reference we could learn a generative model which enforces a certain (distributional) distance to the synthetic reference distribution.

**Conclusion.** In this work, we presented how to improve stance detection models for online political discussions utilizing LLM-generated synthetic data: (i) we showed that fine-tuning with synthetic data related to the question improves the performance of the stance detection model. (ii) We attribute this to the LLM-generated synthetic data aligning well with the real data for the given question, while showing that the BERT model requires data that is more content aligned than structured. (iii) Fine-tuning with synthetic data can be further enhanced by adding the most informative samples which are identified by using the synthetic data as reference. This proves more effective than fine-tuning on all true labels, while using considerably less manually labelled samples, thus reducing labelling effort.

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

# A VISUALIZATIONS

## A.1 VISUALIZING THE SYNTHETIC DATA

We visualize the synthetic data together with the real world data for $M = 1000$ and $M = 200$ in Figures 6 and 7. We plot the data points of the synthetic data in blue and red for the positive and negative samples, respectively. The means of the real world data are plotted as a regular polygon with 8 sides. We observe that the synthetic data extends the real world data, which we consider a factor as to why fine-tuning with synthetic data is effective in online political discussions. Also, the larger the synthetic dataset size, the more the synthetic data matches the distribution of the real world data since for $M = 200$ (see Figure 7) the mean are not as well aligned with the synthetic data. Furthermore, the positive and negative samples are well separated, which we attribute to having pre-trained the BERT-model on $\mathcal{D}_{\text{train}}$ of the X-Stance dataset, giving the prior knowledge about the stance detection task.

## A.2 VISUALIZING THE QUERY STRATEGIES OF THE ACTIVE LEARNING METHODS

We visualize the selected samples of **SQBC**, **CAL** and Random query strategies for $M = 1000$ and $M = 200$ in Figures 8 and 9. We plot the selected samples of the unlabelled data in green. The positive and negative synthetic data samples are plotted in blue and red, respectively. The selected samples are highlighted in orange. We observe that **SQBC** selects the unlabelled samples that are mostly in between the two classes of the synthetic data. This is the expected behaviour since we select the samples where the classification score is ambiguous. For **Random**, the range of selected samples is broad: some similar samples between the two classes like **SQBC** are selected, but also within class samples that are not covered by the synthetic data set. This explains why random selection works well with a large synthetic dataset, since it further extends the decision boundary of the model. For the smaller synthetic dataset $M = 200$, the random selection is not as effective, since the selected samples are spread out over the whole data space and not necessarily in between the two classes as with the larger synthetic dataset. Finally, **CAL** selects samples similar to **SQBC**, but mostly tends to select samples from only one class.

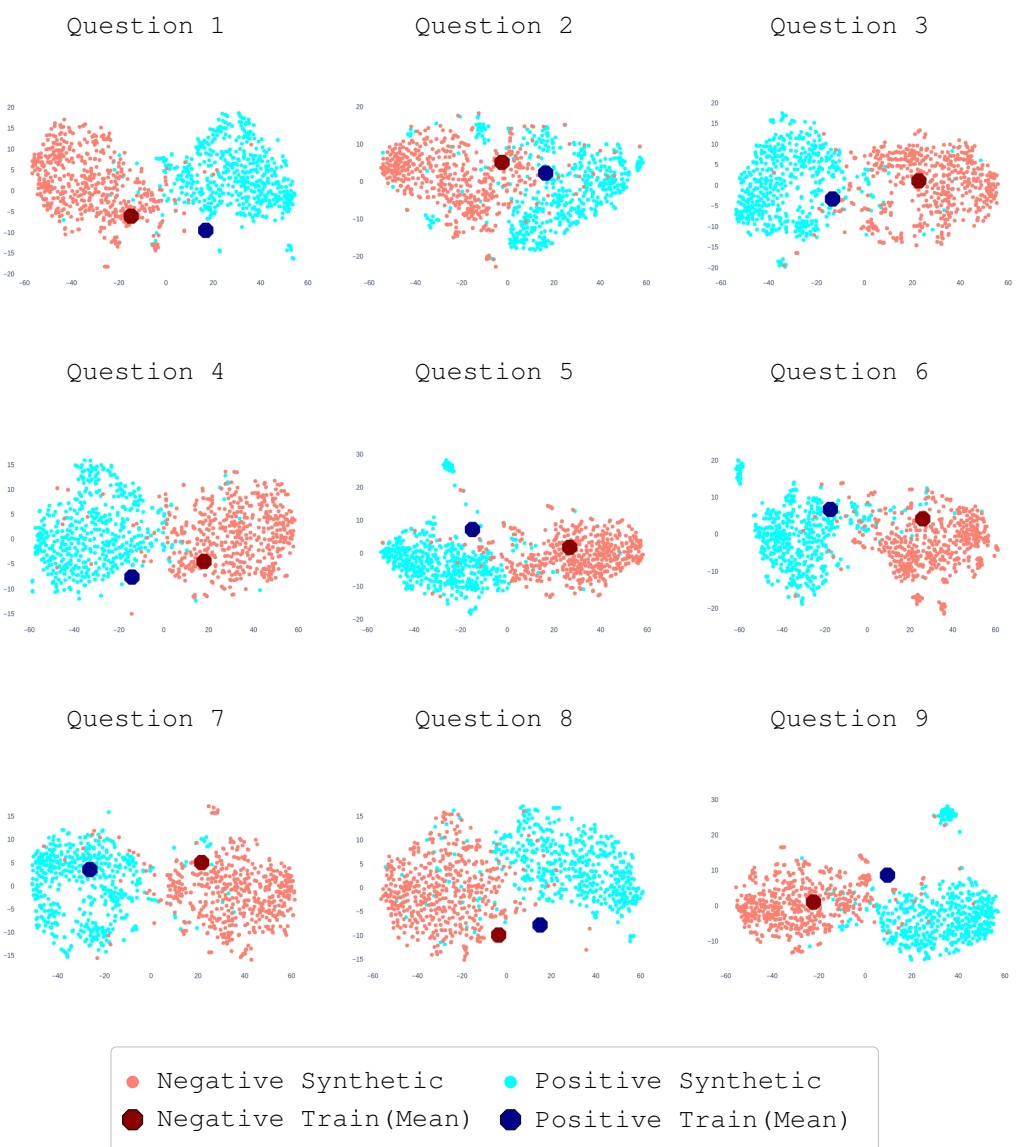

Figure 6: **Visualization of synthetic data with train data means for** $M = 1000$ **synthetic data.** For a larger synthetic dataset size, the means of the synthetic data are well aligned with the real world data and the positive and negative samples are well separated. The synthetic data thus extends the real world data, which we consider a factor as to why fine-tuning with synthetic data is effective in online political discussions.

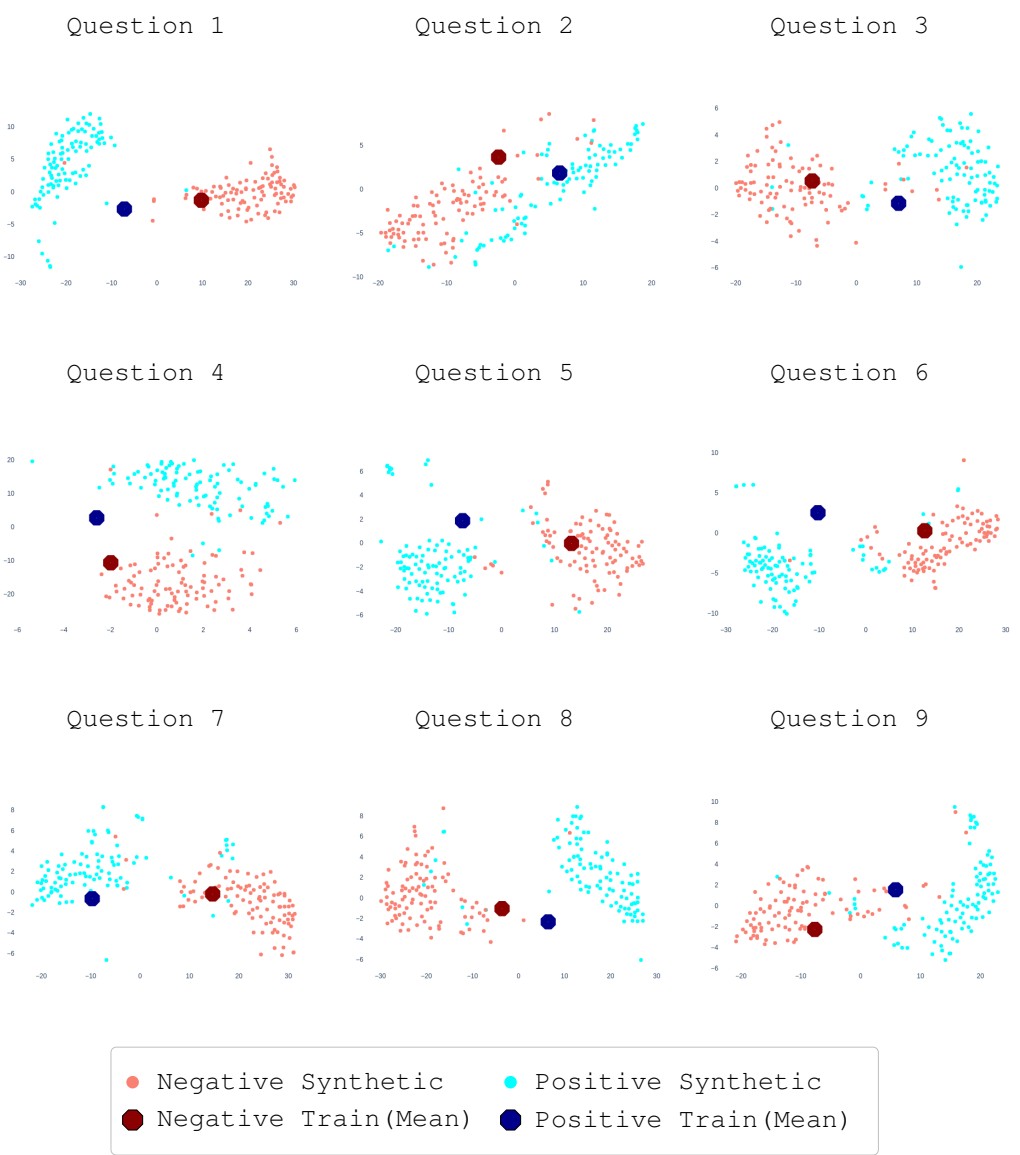

Figure 7: **Visualization of synthetic data with train data means for** $M = 200$ **synthetic data:** For a smaller synthetic dataset size, the means of the synthetic data are not as well aligned with the real world data as for $M = 1000$. However, the positive and negative samples are still well separated.

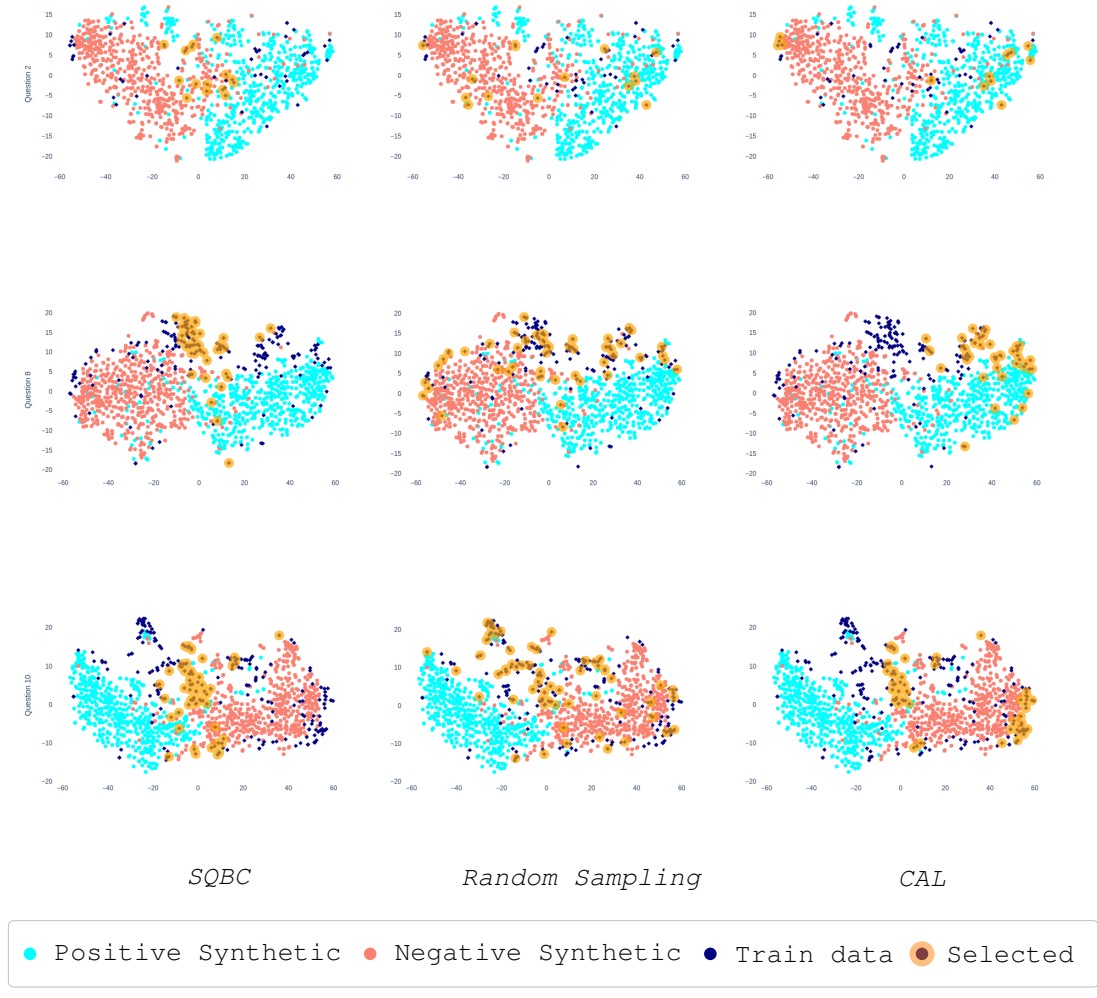

Figure 8: **Visualization of SQBC, Random and CAL query strategies for** $M = 1000$ **synthetic data: SQBC** selects the unlabelled samples that are mostly in between the two classes of the synthetic data. This is the expected behaviour since we select the samples where the classification score is ambiguous. For random selection, the range of selected samples is broad: some similar samples between the two classes like **SQBC** are selected, but also within class samples that are not covered by the synthetic data set. This explains why random selection works well with a large synthetic dataset, since it further extends the decision boundary of the model. Finally, **CAL** selects samples similar to **SQBC**, but mostly tends to select samples from only one class, resulting in worse performance.

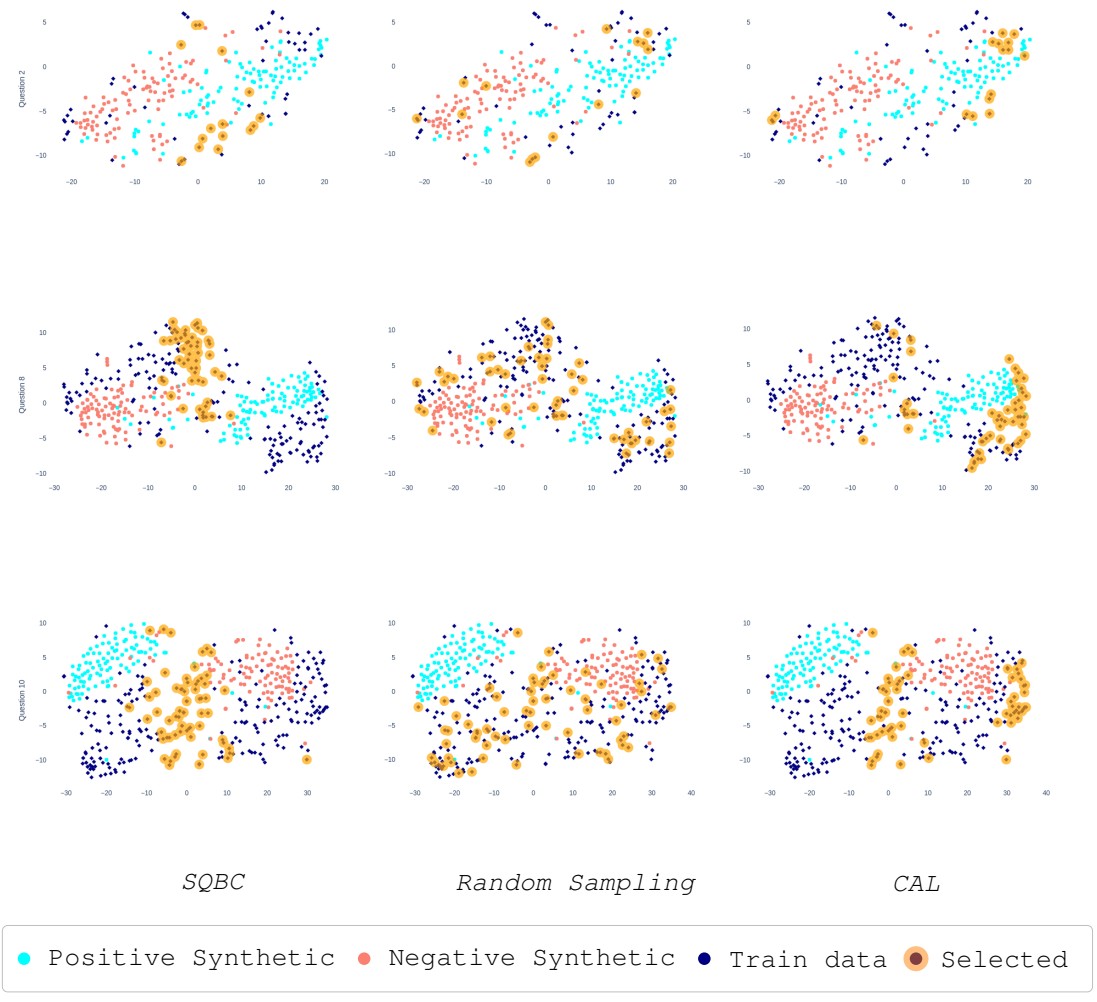

Figure 9: **Visualization of SQBC, Random and CAL query strategies for** $M = 200$ **synthetic data:** For a smaller synthetic dataset size, **SQBC** is still able to select the unlabelled samples that are mostly in between the two classes of the synthetic data. For **Random** we see that the selected samples are a bit further away from the synthetic data distribution, which is why we argue it does not perform as well as with the larger synthetic dataset.

# B   DETAILED RESULTS

| Fine-tuning with synthetic data | | | | |
|---|---|---|---|---|
| | M=0 | M=200 | M=500 | M=1000 |
| Baseline + Synth | 0.693 | 0.712 | 0.718 | 0.723 |
| True Labels + Synth | 0.727 | 0.745 | 0.746 | 0.770 |

Table 4: Tabular version of Figure 2

| | Fine-tuning with most informative samples selected with synthetic data | | | | | | | | | | | |
|---|---|---|---|---|---|---|---|---|---|---|---|---|
| | M=200 | | | | M=500 | | | | M=1000 | | | |
| | 10% | 25% | 50% | 75% | 10% | 25% | 50% | 75% | 10% | 25% | 50% | 75% |
| CAL | 0.693 | 0.697 | 0.705 | 0.714 | 0.692 | 0.694 | 0.707 | 0.718 | 0.692 | 0.696 | 0.708 | 0.720 |
| Random | 0.693 | 0.696 | 0.705 | 0.719 | 0.693 | 0.695 | 0.706 | 0.715 | 0.692 | 0.695 | 0.706 | 0.715 |
| SQBC | 0.693 | 0.697 | 0.709 | 0.722 | 0.692 | 0.700 | 0.711 | 0.722 | 0.692 | 0.698 | 0.712 | 0.721 |

Table 5: Results of only training with most informative samples.

| | Fine-tuning with most informative samples and synthetic data | | | | | | | | | | | |
|---|---|---|---|---|---|---|---|---|---|---|---|---|
| | M=200 | | | | M=500 | | | | M=1000 | | | |
| | 10% | 25% | 50% | 75% | 10% | 25% | 50% | 75% | 10% | 25% | 50% | 75% |
| CAL+Synth | 0.713 | 0.715 | 0.727 | 0.732 | 0.711 | 0.721 | 0.732 | 0.748 | 0.695 | 0.715 | 0.747 | 0.749 |
| Random+Synth | 0.716 | 0.720 | 0.724 | 0.734 | 0.723 | 0.735 | 0.730 | 0.748 | 0.724 | 0.746 | 0.754 | 0.756 |
| SQBC+Synth | 0.715 | 0.723 | 0.731 | 0.735 | 0.714 | 0.726 | 0.744 | 0.750 | 0.721 | 0.737 | 0.753 | 0.747 |

Table 6: Tabular version of Figure 4

## C  SYNTHETIC DATA

### C.1  ANALYSING THE SYNTHETIC DATA DISTRIBUTION

As described in Section 5.2, we analyse the distribution of the synthetic data in comparison to the real world data. In the following tables we provide samples of the interquartile ranges of the entropy distribution. Tables C.1 and C.1 contain the same samples in English and German respectively. Table C.1 contain comments from the real data distribution.

We see that that the synthetic data samples are longer than the real data samples and contain more reserved language and practically no emotional language. In comparison, the real data samples are shorter and concise with some samples in the lower entropy range containing more emotional language or little information. This however is common for online political discussions. In any case, we see that the underlying notions behind the comments are similar in both the synthetic and real data samples.

| Distribution Range | Comment |
| --- | --- |
| Minimum | Sure, I can help you with that! Please write your message and I will assist you with it. |
| $0\% - 25\%$ | I strongly disagree with the idea that the residence permit for migrants from non-EU/EFTA countries should be linked to the fulfillment of binding integration agreements throughout Switzerland. While it's important that migrants integrate into Swiss society and contribute to the country, I believe that it unfair to tie their ability to live and work in Switzerland to their compliance with certain conditions. It could lead to discrimination against some migrants who may struggle to meet these requirements or face cultural barriers. Instead, I propose a more nuanced approach that takes into account the unique circumstances and needs of each individual migrant, while still promoting integration and equality for all. |
| $25\% - 50\%$ | As someone who strongly believes in the importance of immigration policies that promote successful integration and contribution to society, I fully support the idea of linking residence permits for migrants from non-EU/EFTA countries to the fulfillment of binding integration agreements throughout Switzerland. This approach ensures that migrants committed to making Switzerland their home and contributing positively to the country's well-being can enjoy full legal recognition and protection while being held accountable for fulfilling their responsibilities. By creating an incentive system where residency and benefits are directly tied to compliance with integration agreements, we can foster a culture of shared responsibility and mutual benefit between incoming migrants and Swiss society as a whole. |
| $50\% - 75\%$ | Sure, I can help you generate a response for this scenario. From your perspective, you might write something like: I strongly believe that the residence permit for migrants from non-EU/EFTA countries should be linked to the fulfillment of binding integration agreements throughout Switzerland. As someone who values the diversity and richness that migration brings to our country, I think it's important that we have clear expectations and requirements for those coming here to live and work. Integration agreements should include things like language proficiency, cultural awareness, and contributions to community life. By linking these agreements to the residency permit, we can ensure that migrants are fully committed to integrating into Swiss society and contributing to its well-being. Of course, there may be some concerns or challenges with this approach, but I believe that the benefits far outweigh any potential drawbacks. A strong and diverse immigration policy is crucial for the future of Switzerland, and by setting high standards for integration, we can attract the best and brightest from around the world. |
| $75\% - 100\%$ | I believe that requiring residence permits for migrants from non-EU/EFTA countries to be linked to the fulfillment of binding integration agreements throughout Switzerland is a fair and necessary policy. By doing so, we can ensure that migrants who come to Switzerland have a clear understanding of their responsibilities and the requirements they need to meet in order to live and work here. This also helps to promote a sense of community and integration among all residents of Switzerland, regardless of their place of birth or nationality. Additionally, it allows us to carefully screen and select immigrants who are likely to be successful and positive contributing members of our society. Overall, I believe that this policy will benefit both migrants and Swiss citizens alike by promoting mutual respect and understanding. |
| Maximum | I would argue that linking the residence permit for migrants from non-EU/EFTA countries to the fulfillment of mandatory integration agreements across Switzerland is too burdensome and unreasonable. This creates an unfair situation in which people who come to Switzerland to work or study are essentially forced to make a long-term commitment before they even have access to the legal rights they need to live and work here comfortably.The Commission has asked the Commission to review the conditions set out in Regulation (EC) No. 1225/2009 for the application of Regulation (EC) No. 1225/2009 of the Council concerning the application of Regulation (EC) No. 1225/2009 of the Council concerning the application of Regulation (EC) No. 1225/2009 of the Council concerning the application of Regulation (EC) No. 1225/2009 of the Council concerning the application of Regulation (EC) No. 1225/2009 of the Council concerning the application of Regulation (EC) No. 1225/2009 of the Council concerning the application of Regulation (EC) No. 1225/2009 of the Council concerning the application of Regulation (EC) No. 1225/2009 of the Council concerning the application of Regulation (EC) No. 1225/2009 of the Council concerning the application of Regulation (EC) No. 1225/2009 of the Council concerning the application of Regulation (EC) No. 1225/2009 of the Council concerning the application of Regulation (EC) No. 1225/2009 of the Council concerning the application of Regulation (EC) No. 1225/2009 of the Council concerning the application of Regulation (EC) No. 1225/2009 of the Council concerning the application of Regulation (EC) No.I believe that the government should instead focus on creating more flexible and streamlined processes to facilitate the integration of migrants into Swiss society without imposing unnecessary restrictions or requirements. |

Table 7: English comments sampled from the interquartire ranges of the entropy distribution of the **synthetic** data.

| Distribution Range | Comment |
| --- | --- |
| Minimum | Schreiben Sie Ihre Nachricht, ich helfe Ihnen. |
| $0\% - 25\%$ | Ich bin mit der Vorstellung, dass die Aufenthaltserlaubnis für Migranten aus Nicht-EU/EFTA-Ländern an die Erfüllung verbindlicher Integrationsvereinbarungen in der gesamten Schweiz gebunden werden sollte, voll und ganz nicht einverstanden. Es ist zwar wichtig, dass Migranten in die Schweizer Gesellschaft integriert werden und einen Beitrag für das Land leisten, aber ich glaube, es ist unfair, ihre Fähigkeit, in der Schweiz zu leben und zu arbeiten, an die Einhaltung bestimmter Bedingungen zu knüpfen. Es könnte zu Diskrimminierung einiger Migranten führen, die möglicherweise Schwierigkeiten haben, diese Anforderungen zu erfüllen oder kulturellen Hindernissen ausgesetzt sind. Ich schlage stattdessen einen differenzierteren Ansatz vor, der die einzigartigen Umstände und Bedürfnisse jedes einzelnen Migranten berücksichtigt und gleichzeitig Integration und Gleichberechtigung für alle Migranten fördert. |
| $25\% - 50\%$ | Als jemand, der fest an die Bedeutung von Einwanderungspolitiken glaubt, die eine erfolgreiche Integration und einen Beitrag zur Gesellschaft fördern, unterstütze ich die Idee, Aufenthaltsgenehmigungen für Migranten aus Nicht-EU/EFTA-Ländern mit der Erfüllung verbindlicher Integrationsvereinbarungen in der gesamten Schweiz zu verknüpfen. Dieser Ansatz gewährleistet, dass Migranten, die sich verpflichtet haben, die Schweiz zu ihrer Heimat zu machen und einen positiven Beitrag zum Wohlergehen des Landes leisten, vollständige rechtliche Anerkennung und Schutz genießen können, während sie gleichzeitig für die Erfüllung ihrer Pflichten zur Rechenschaft gezogen werden. Durch die Schaffung eines Anreizsystems, bei dem Aufenthalt und Leistungen unmittelbar an die Einhaltung der Integrationsvereinbarungen gebunden sind, können wir eine Kultur der gemeinsamen Verantwortung und des gegenseitigen Nutzens zwischen den einwandernden Migranten und der Schweizer Gesellschaft als Ganzes fördern. |
| $50\% - 75\%$ | Sicher, ich kann Ihnen helfen, eine Antwort für dieses Szenario zu generieren Aus Ihrer Sicht könnten Sie etwa schreiben: Ich bin der festen Überzeugung, dass die Aufenthaltserlaubnis für Migranten aus Nicht-EU/EFTA-Ländern mit der Erfüllung verbindlicher Integrationsvereinbarungen in der gesamten Schweiz verbunden sein sollte Als jemand, der die Vielfalt und den Reichtum schätzt, die Migration unserem Land bringt, denke ich, dass es wichtig ist, dass wir klare Erwartungen und Anforderungen an diejenigen haben, die hierher kommen, um zu leben und zu arbeiten. Die Integrationsvereinbarungen sollten Aspekte wie Sprachkenntnisse, kulturelles Bewusstsein und Beiträge zum Gemeinschaftsleben umfassen Durch die Verknüpfung dieser Vereinbarungen mit der Aufenthaltserlaubnis können wir sicherstellen, dass Migranten sich voll und ganz der Integration in die Schweizer Gesellschaft und dem Wohlergehen der Schweizer Gesellschaft verschrieben haben. Natürlich kann dieser Ansatz einige Bedenken oder Herausforderungen mit sich bringen, aber ich glaube, dass die Vorteile weit überwiegen alle möglichen Nachteile Eine starke und vielfältige Einwanderungspolitik ist für die Zukunft der Schweiz von entscheidender Bedeutung. Durch die Festlegung hoher Integrationsstandards können wir die Besten und Klügsten aus der ganzen Welt anziehen. |
| $75\% - 100\%$ | Ich glaube, dass die Anerkennung der Aufenthaltsgenehmigung für Migranten aus Nicht-EU/EFTA-Ländern an die Erfüllung verbindlicher Integrationsvereinbarungen in der gesamten Schweiz geknüpft ist, eine faire und notwendige Politik Auf diese Weise können wir sicherstellen, dass Migranten, die in die Schweiz kommen, ein klares Verständnis ihrer Verantwortlichkeiten und der Anforderungen haben, die sie erfüllen müssen, um hier zu leben und zu arbeiten. Dies trägt auch dazu bei, ein Gefühl der Gemeinschaft und Integration unter allen Einwohnern der Schweiz zu fördern, unabhängig von ihrem Geburtsort oder ihrer Staatsangehörigkeit. Darüber hinaus können wir so Migranten sorgfältig auswählen, die wahrscheinlich erfolgreiche und positive Mitglieder unserer Gesellschaft sind. Ich glaube, dass diese Politik sowohl den Migranten als auch den Schweizer Bürgern zugute kommen wird, indem sie gegenseitigen Respekt und Verständnis fördert |
| Maximum | Ich würde argumentieren, dass die Verknüpfung der Aufenthaltserlaubnis für Migranten aus Nicht-EU/EFTA-Ländern mit der Erfüllung verbindlicher Integrationsvereinbarungen in der gesamten Schweiz zu belastend und unvernünftig ist Es entsteht eine ungerechte Situation, in der Menschen, die zur Arbeit oder zum Studium in die Schweiz kommen, im Wesentlichen gezwungen sind, eine langfristige Verpflichtung einzugehen, bevor sie überhaupt Zugang zu den gesetzlichen Rechten haben, die sie benötigen, um hier bequem zu leben und zu arbeiten. Die Kommission hat die Kommission aufgefordert, die in der Verordnung (EG) Nr. 1225/2009 festgelegten Bedingungen für die Anwendung der Verordnung (EG) Nr. 1225/2009 des Rates über die Anwendung der Verordnung (EG) Nr. 1225/2009 des Rates über die Anwendung der Verordnung (EG) Nr. 1225/2009 des Rates über die Anwendung der Verordnung (EG) Nr. 1225/2009 des Rates über die Anwendung der Verordnung (EG) Nr. 1225/2009 des Rates über die Anwendung der Verordnung (EG) Nr. 1225/2009 des Rates über die Anwendung der Verordnung (EG) Nr. 1225/2009 des Rates über die Anwendung der Verordnung (EG) Nr. 1225/2009 des Rates über die Anwendung der Verordnung (EG) Nr. 1225/2009 des Rates über die Anwendung der Verordnung (EG) Nr. 1225/2009 des Rates über die Anwendung der Verordnung (EG) Nr. 1225/2009 des Rates über die Anwendung der Verordnung (EG) Nr. 1225/2009 des Rates über die Anwendung der Verordnung (EG) Nr. 1225/2009 des Rates über die Anwendung der Verordnung (EG) Nr. Ich glaube, dass die Regierung sich stattdessen auf die Schaffung flexiblerer und schlankerer Prozesse konzentrieren sollte, um die Integration von Migranten in die Schweizer Gesellschaft zu erleichtern, ohne unnötige Einschränkungen oder Anforderungen aufzuerlegen. |

Table 8: Corresponding German comments sampled from the interquartire ranges of the entropy distribution of the **synthetic** data.

| Distribution Range | Comment |
|---|---|
| Minimum | muss die integrationsvereinbarungen noch studieren |
| $0\% - 25\%$ | Wer sich anpasst und korrekt verhaltet darf auch hier bleiben. |
| $25\% - 50\%$ | Integrationsvereinbarungen nicht generell verhängen, sondern individuell anordnen, wenn die Person auch nach längerer Anwesenheit in der Schweiz Integrationsdefizite hat oder bei neu zugezogenen Personen, wenn mit Anpassungsschwierigkeiten zu rechnen ist. Keine Standardformulare, keine Automatismen. |
| $50\% - 75\%$ | Ich zweifle an der Umsetzbarkeit solcher Methoden, eines "Vertrags". Es ist anstrebenswert, dass ein guter Integrationsablauf statt finden kann. Allerdings ist dafür immer eine Gesellschaft notwendig, die Personen aufnehmen und integrieren will, wie auch Personen die diese Möglichkeit wahrnehmen können und wollen. |
| $75\% - 100\%$ | Ich finde die Idee Integrationsmassnahmen verbindlich zu machen sinnvoll. Sprachkurse sind ein ideales Instrument dafür. Es darf aber nicht sein, dass gewisse Forderungen zur Voraussetzung werden für die Aufenthaltserlaubnis. Nicht jeder Mensch lernt gleich schnell eine Sprache oder kann sich gleich schnell integrieren. Missbräuche und Diskriminierungen können auftreten. Dies muss verhindert werden. Ich wäre eher für einen obligatorischen Kostenlosen-Sprachkurs. |
| Maximum | Heikler Punkt, im Prinzip JA für Alle, nur - auch gleiche Rechte/Pflichten für alle - warum sollen die EU-EFTA-Bürger keine solchen Integrationsvereinbarungen abschliessen. Der bildungsferne Rumäne fährt besser als der hochausgebildete Chinese/US-Amerikaner/IT-Spezialist aus Indien. Die ganze Einwanderung muss im Sinne der MEI umgesetzt werden. |

Table 9: Corresponding German comments sampled from the interquartire ranges of the entropy distribution of the **real** data.

## C.2 TRANSLATED DATA SAMPLES

We show the translated questions used for synthetic data generation in Table 10 and some samples of generated comments in 12. We see that the questions are translated correctly and synthetic data can be generated for both favor and against stances.

| Question in German | Question in English |
|---|---|
| Sollen sich die Versicherten stärker an den Gesundheitskosten beteiligen (z.B. Erhöhung der Mindestfranchise) | Should insured persons contribute more to health costs (e.g. increase in the minimum deductible)? |
| Befürworten Sie ein generelles Werbeverbot für Alkohol und Tabak? | Do you support a general ban on advertising alcohol and tobacco? |
| Soll eine Impfpflicht für Kinder gemäss dem schweizerischen Impfplan eingeführt werden? | Should compulsory vaccination of children be introduced in accordance with the Swiss vaccination schedule? |
| Soll die Aufenthaltserlaubnis für Migrant/innen aus Nicht-EU/EFTA-Staaten schweizweit an die Erfüllung verbindlicher Integrationsvereinbarungen geknüpft werden? | Should the residence permit for migrants from non-EU/EFTA countries be linked to the fulfilment of binding integration agreements throughout Switzerland? |
| Soll der Bund erneuerbare Energien stärker fördern? | Should the federal government promote renewable energy more? |
| Befürworten Sie eine strengere Kontrolle der Lohngleichheit von Frauen und Männern? | Do you support stricter control of gender pay? |
| Soll der Bund die finanzielle Unterstützung für die berufliche Weiterbildung und Umschulung ausbauen? | Should the Federal Government expand its financial support for continuing vocational training and retraining? |
| Sollen in der Schweiz vermehrt Spitäler geschlossen werden, um die Kosten im Gesundheitsbereich zu senken? | Should more hospitals be closed in Switzerland to reduce healthcare costs? |
| Eine eidenössische Volksinitiative verlangt, dass der Bundesrat direkt vom Volk gewählt werden soll. Unterstützen Sie dieses Anliegen? | A sworn federal people's initiative demands that the Federal Council be elected directly by the people. Do you support this concern? |
| Soll die Einführung der elektronischen Stimmabgabe bei Wahlen und Abstimmungen (E-Voting) weiter vorangetrieben werden? | Should the introduction of electronic voting in elections and voting (e-voting) be further promoted? |

Table 10: Chosen questions for stance detection in German and their English translation

| Comment in English | Comment in German |
|---|---|
| I don't believe that insured persons should contribute more to their own healthcare costs by increasing the minimum deductible. While it may seem like a good idea to increase deductibles as a way to reduce overall healthcare costs, this only shifts the burden onto patients and puts them at risk of facing high out-of-pocket expenses. It's important for insurance companies to find ways to control costs without placing the onus on insured individuals. | Ich glaube nicht, dass Versicherte mehr zu ihren eigenen Gesundheitskosten beitragen sollten, indem sie den Mindestabzug erhöhen. Die Erhöhung der Selbstbehalte als Mittel zur Senkung der gesamten Gesundheitskosten mag zwar als gute Idee erscheinen, verschiebt jedoch die Belastung nur auf die Patienten und gefährdet sie mit hohen Kosten, die sie selbst tragen müssen. Es ist wichtig für Versicherungsgesellschaften, Wege zu finden, Kosten zu kontrollieren, ohne die Versicherten mit der Last zu belasten. |
| Hi, friends! I think it's a great idea for insured person to contribute more to their health costs by increasing the minimum deductible. As someone who believes in taking ownership of my own health and finances, I believe that paying more out-of-pocket can help me better manage my expenses and make more informed decisions about my healthcare. Plus, I think that by contributing more, I can save money in the long run and avoid potential financial strain in other areas of my life. | Ich finde es eine gute Idee, dass Versicherte mehr zu ihren Gesundheitskosten beitragen, indem sie den Mindestabzug erhöhen. Als jemand, der daran glaubt, die Verantwortung für seine Gesundheit und seine Finanzen zu übernehmen, glaube ich, dass mehr Geld aus eigener Tasche helfen kann, meine Ausgaben besser zu managen und fundierte Entscheidungen über meine Gesundheitsversorgung zu treffen. Außerdem denke ich, dass ich durch einen größeren Beitrag auf lange Sicht Geld sparen und finanzielle Belastungen in anderen Bereichen meines Lebens vermeiden kann. |
| "In my opinion, it's wrong to link the residence permit for migrants from non-EU/EFTA countries to the fulfillment of binding integration agreements in Switzerland. This policy puts too much emphasis on the ability of migrants to adapt to Swiss culture and language at the expense of their personal freedoms and rights as human beings. It also creates incentives for migrants to cheat or lie in order to gain a residency permit and access to valuable services and resources." | "Meiner Meinung nach ist es falsch, die Aufenthaltserlaubnis für Migranten aus Nicht-EU/EFTA-Ländern mit der Erfüllung verbindlicher Integrationsvereinbarungen in der Schweiz zu verbinden. Diese Politik legt zu viel Wert auf die Fähigkeit der Migranten, sich an die Schweizer Kultur und Sprache anzupassen, auf Kosten ihrer persönlichen Freiheiten und Rechte als Menschen. Es schafft auch Anreize für Migranten, zu betrügen oder zu lügen, um eine Aufenthaltserlaubnis zu erhalten und Zugang zu wertvollen Dienstleistungen und Ressourcen zu erhalten " |
| I believe that the residence permit for migrants from non-EU/EFTA countries should be linked to the fulfillment of binding integration agreements throughout Switzerland. This is because integration should not be seen as a one-sided process, but rather as a mutually beneficial relationship between the immigrant and the host country. By linking the residence permit to the successful completion of integration agreements, we incentivize immigrants to actively participate in the integration process and become active members of Swiss society. Additionally, this approach ensures that immigrants do not receive a free ride by simply obtaining a residence permit without contributing to the well-being of Swiss society. Ultimately, a comprehensive integration approach benefits both the immigrant and the host country by creating a more inclusive and harmonious society. | Ich glaube, dass die Aufenthaltserlaubnis für Migranten aus Nicht-EU/EFTA-Ländern mit der Erfüllung verbindlicher Integrationsvereinbarungen in der gesamten Schweiz verbunden sein sollte. Die Integration sollte nicht als einseitiger Prozess, sondern als eine gegenseitig vorteilhafte Beziehung zwischen dem Einwanderer und dem Aufnahmeland betrachtet werden. Durch die Verknüpfung der Aufenthaltserlaubnis mit dem erfolgreichen Abschluss von Integrationsvereinbarungen fördern wir die aktive Teilnahme der Einwanderer am Integrationsprozess und die Förderung ihrer Teilnahme an der Schweizer Gesellschaft Darüber hinaus wird durch diese Vorgehensweise sichergestellt, dass Einwanderer nicht einfach eine Aufenthaltserlaubnis erhalten, ohne zum Wohlergehen der Schweizer Gesellschaft beizutragen. Letztlich kommt einem umfassenden Integrationsansatz sowohl der Einwanderer als auch dem Aufnahmeland zugute, da er eine integrativere und harmonischere Gesellschaft schafft |

Table 11: Sample of translated comments from comments generated by the LLM used for fine-tuning the stance detection model.

## C.3 SYNTHETIC DATA STANCE ALIGNMENT

| Comment in English | Intended Stance |
|---|---|
| I don't believe that insured persons should contribute more to their own healthcare costs by increasing the minimum deductible. While it may seem like a good idea to increase deductibles as a way to reduce overall healthcare costs, this only shifts the burden onto patients and puts them at risk of facing high out-of-pocket expenses. It's important for insurance companies to find ways to control costs without placing the onus on insured individuals. | AGAINST |
| Hi, friends! I think it's a great idea for insured person to contribute more to their health costs by increasing the minimum deductible. As someone who believes in taking ownership of my own health and finances, I believe that paying more out-of-pocket can help me better manage my expenses and make more informed decisions about my healthcare. Plus, I think that by contributing more, I can save money in the long run and avoid potential financial strain in other areas of my life. | FAVOR |
| "In my opinion, it's wrong to link the residence permit for migrants from non-EU/EFTA countries to the fulfillment of binding integration agreements in Switzerland. This policy puts too much emphasis on the ability of migrants to adapt to Swiss culture and language at the expense of their personal freedoms and rights as human beings. It also creates incentives for migrants to cheat or lie in order to gain a residency permit and access to valuable services and resources." | AGAINST |
| I believe that the residence permit for migrants from non-EU/EFTA countries should be linked to the fulfillment of binding integration agreements throughout Switzerland. This is because integration should not be seen as a one-sided process, but rather as a mutually beneficial relationship between the immigrant and the host country. By linking the residence permit to the successful completion of integration agreements, we incentivize immigrants to actively participate in the integration process and become active members of Swiss society. Additionally, this approach ensures that immigrants do not receive a free ride by simply obtaining a residence permit without contributing to the well-being of Swiss society. Ultimately, a comprehensive integration approach benefits both the immigrant and the host country by creating a more inclusive and harmonious society. | FAVOR |
| As someone who is against stricter controls on equal pay for women and men, I believe that all individuals should be paid based on their individual skills, performance, and contributions to the company, regardless of their gender. I think it's important to recognize and reward hard work and dedication, regardless of whether someone is male or female. Additionally, I argue that it's already illegal and unethical to discriminate on the basis of gender and that such regulations would only create unnecessary tensions and potential conflicts within the workplace. Instead, we should focus on creating a truly equitable society where everyone has access to equal opportunities for education, job training, and development. | AGAINST |
| I personally believe that stricter controls on equal pay for women and men are necessary because it's not always fair that men receive higher salaries just because of their gender. In many cases, women have to work harder than men to achieve the same level of performance, but they still earn less money. Stricter controls would help ensure that everyone's hard work is recognized and rewarded fairly, regardless of their gender. Additionally, it's important to promote gender equality and give all individuals an equal opportunity to succeed. Overall, I think stricter controls on equal pay are necessary to help reduce the gender pay gap and create a more equitable society. | FAVOR |

Table 12: Sample of comments generated by the LLM used for fine-tuning the stance detection model.

# D    ADDITIONAL EXPERIMENTAL DETAILS

## D.1    EVALUATION.

For fine-tuning and testing we evaluate the given model separately on 10 chosen questions from the test dataset of X-Stance for all experiments. For each question $q$ we split $\mathcal{D}_{\text{test}}^{(q)}$ into a $60/40$ train/test split (repeated with 5 different seeds to get error bars) and use the train split for fine-tuning to the given question and the test split for evaluation. Our main results report the average F1 score over 10 selected questions from the test dataset evaluated on the comments from the test split. The error bars represent the average standard deviation over the 10 questions for 5 runs with different seeds. More detailed results per question are shown in Appendix F.

## D.2    COMPUTE AND RUNTIME

We conduct our experiments on a single NVIDIA A100 80GB GPU and a 32 core CPU. With this setup, for Mistral-7B, the generation of synthetic data takes approximately 3 hours per question for a synthetic dataset size of $M = 1000$. Fine-tuning the BERT model with the synthetic data takes less than a minute. For the active learning methods, the selection of the most informative samples takes less than a minute. Hence the largest computational effort is the generation of the synthetic data.

## D.3    TRANSLATION OF THE X-STANCE DATASET FOR SYNTHETIC DATA GENERATION

In Figure 10 we show the pipeline for translating the X-Stance dataset for synthetic data generation. We start with a question $q$ from the X-Stance test dataset and translate the question to English with a NLLB-330M model (NLLB Team et al., 2022). Then we let the Mistral-7B model generate synthetic data, i.e., comments for the translated question. The generated comments are then translated back to German to be used for fine-tuning the model in our experiments.

## D.4    OVERVIEW OF USED DATASETS

In Table 13 we show an overview of the datasets used in our experiments for the different methods we evaluate.

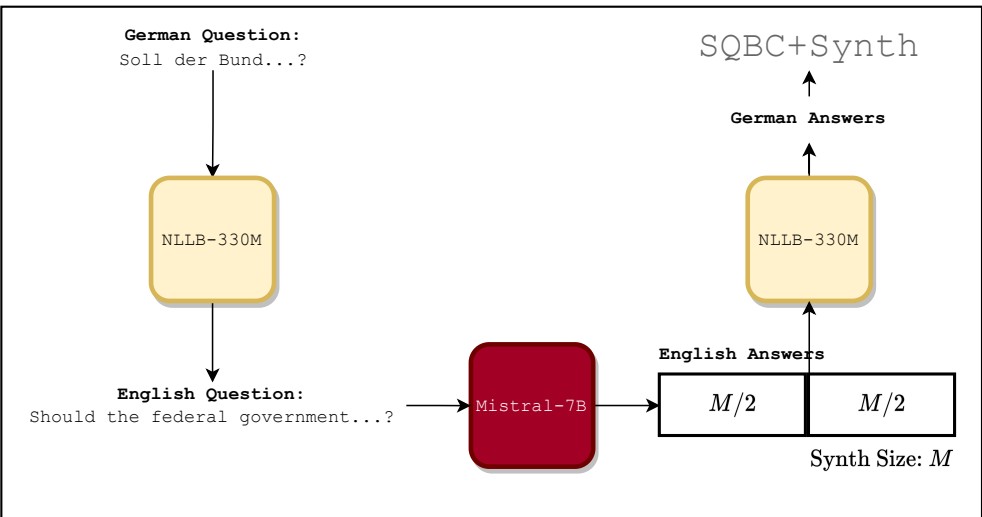

Figure 10: **Overview of the pipeline for active learning with synthetic data:** We start with a question $q$ from the X-Stance test dataset and translate the question to English with a NLLB-330M model (NLLB Team et al., 2022). Then we let the Mistral-7B model generate synthetic data, i.e., comments for the translated question. The generated comments are then translated back to German to be used for fine-tuning the model in our experiments.

| Config \ Datasets | Manual labels $D_{\text{MInf}}$ | True Labels $\mathcal{D}_t$ | Synth Aug $D_{\text{synth}}$ |
|---|---|---|---|
| Baseline | | | |
| Baseline + Synth | | | ✓ |
| True Labels | | ✓ | |
| True Labels + Synth | | ✓ | ✓ |
| SQBC | ✓ | | |
| SQBC + Synth | ✓ | | ✓ |
| CAL | ✓ | | |
| CAL + Synth | ✓ | | ✓ |
| Random | ✓ (randomly selected) | | |
| Random + Synth | ✓ (randomly selected) | | ✓ |

Table 13: Synth: Synthetic Data, Aug: Augmentation. We compare different variants of active learning with synthetic data.

# E  DATASET

X-Stance is a multilingual stance detection dataset, including comments in German ($48,600$), French ($17,200$) and Italian ($1,400$) on political questions, answered by election candidates from Switzerland. The data has been extracted from smartvote[1], a Swiss voting advice platform. For the task of cross-topic stance detection the data is split into a training set, including questions on 10 political topics, and a test set with questions on two topics that have been held out, namely *healthcare* and *political system*.

## E.1  CHOSEN QUESTIONS AND THEIR DISTRIBUTION

We present the 10 chosen questions for our experiments in Table 10. We show the original questions in German and their corresponding English translations by the translation model. Furthermore, we also show the (60 / 40 ) train/test split for each question in Figure 11. We chose 10 questions that reflect the overall distribution of $\mathcal{D}_{\text{test}}^{(q)}$. We choose questions with small amount of comments, unbalanced comments and also balanced comments. Furthermore, for 5 of the questions the majority class is *favor* and for the other 5 the majority class is *against*.

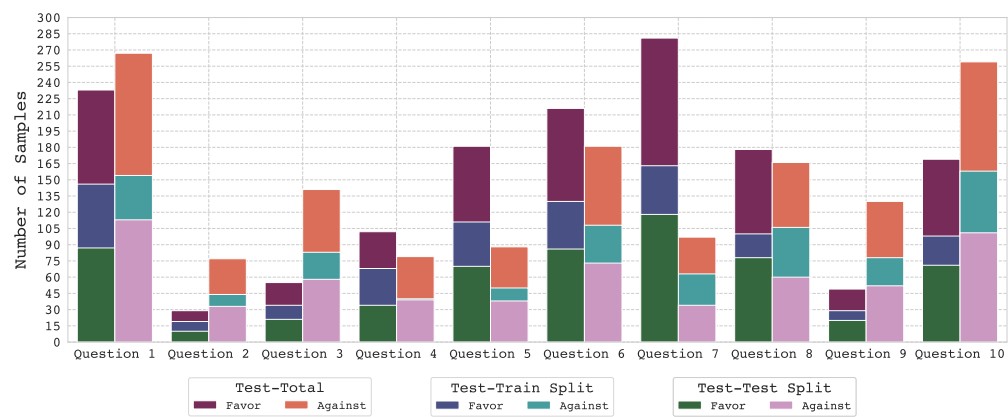

Figure 11: **Distribution of the positive and negative samples for the train and test split of $\mathcal{D}_{\text{test}}^{(q)}$:** We show the distribution of the positive and negative samples of the X-Stance test dataset for the questions Q1-Q10. We also show the 60/40 train/test split for the 10 questions. We chose 10 questions that reflect the overall distribution of $\mathcal{D}_{\text{test}}^{(q)}$. We chose unbalanced, balanced and low sample size questions to evaluate the effectiveness of our approach.

---

[1] https://www.smartvote.ch/

## F    EXTENDED RESULTS

We present the extended results for the different synthetic dataset sizes $M = 200$, $M = 500$ and $M = 1000$ in Figures 12, 13 and 14. As in Figure 4, we show the results for the different active learning methods and the different configurations of the synthetic data, while varying the amount of samples that need to be labelled. We compare all methods to **True Labels**, hence the horizontal line corresponds to the performance of the baseline model fine-tuned with the true labels.

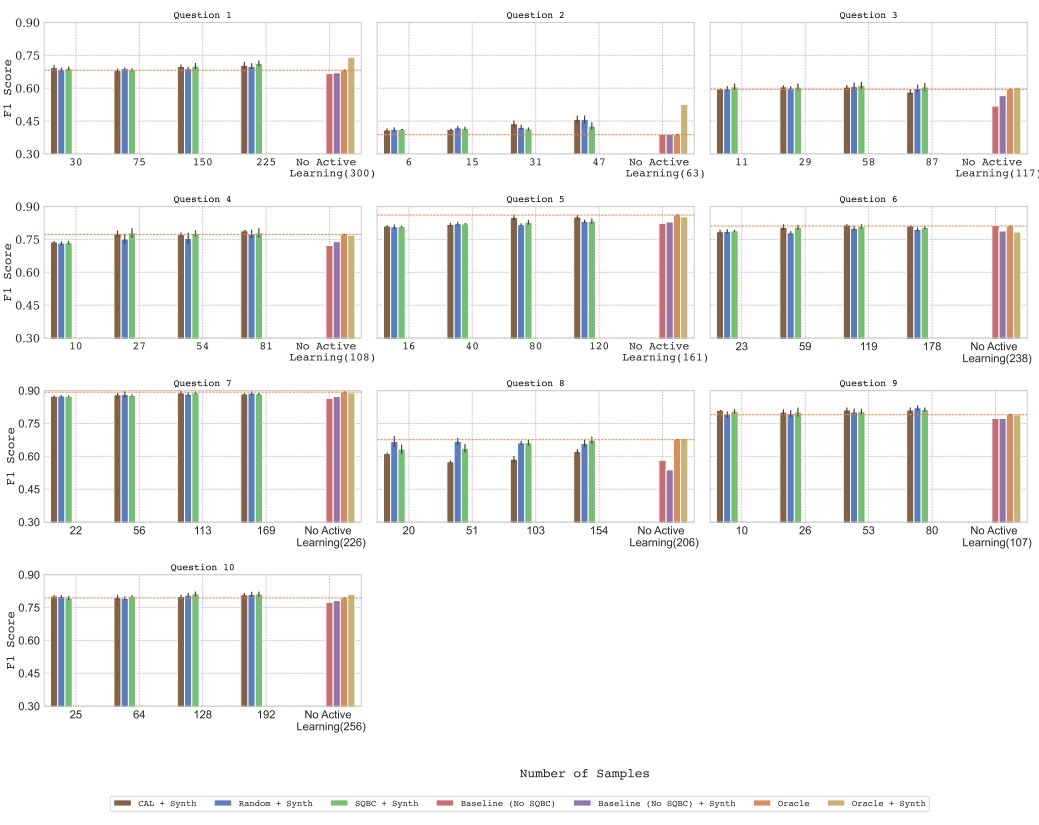

Figure 12: **Extended results of Figure 4 for M=200:**

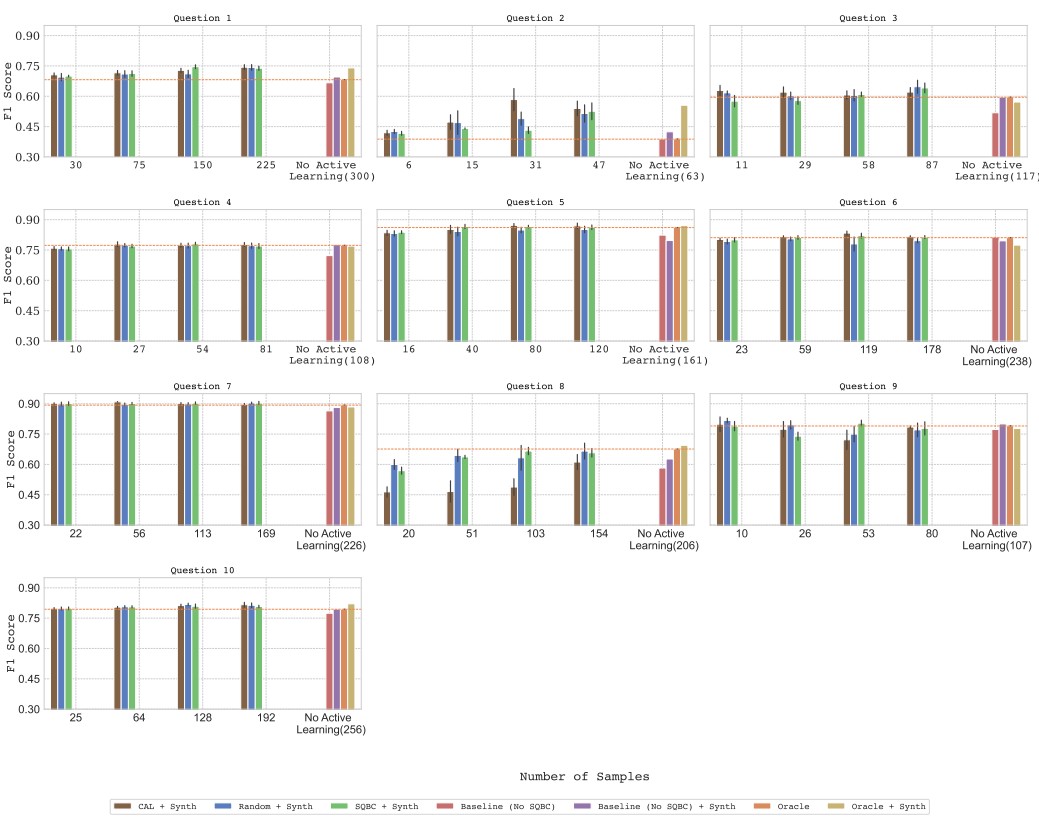

Figure 13: **Extended results of Figure 4 for M=500**

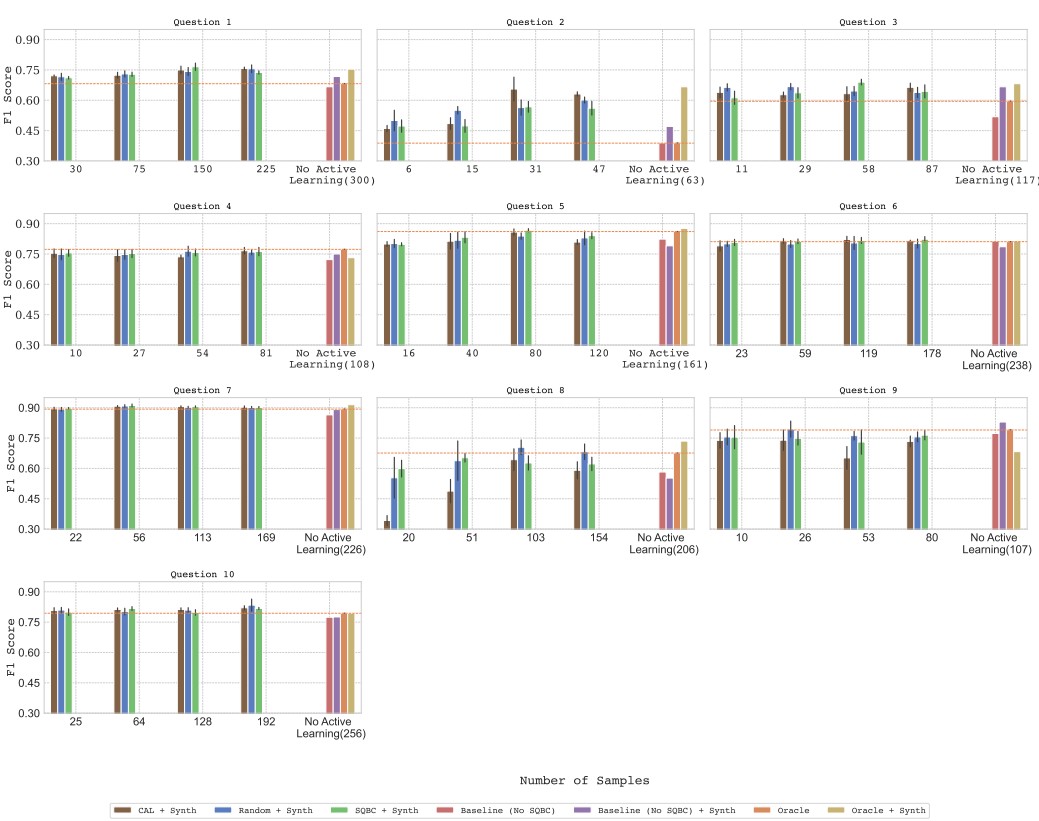

Figure 14: **Extended results of Figure 4 for M=1000**

