# OpenReview forum: "The Power of LLM-Generated Synthetic Data for Stance Detection in Online Political Discussions"
_ICLR.cc/2025/Conference — ICLR 2025 Spotlight_

### Official Review · Reviewer_si6z · 2024-11-02

**Soundness:** 2
**Presentation:** 3
**Contribution:** 3
**Rating:** 8
**Confidence:** 3

**Summary:**

Yet another synthetic data paper that shows modest improvements but doesn't quite nail why or how to make synthetic data actually useful. Some interesting ideas buried under conventional methodology.

Look, I've seen enough "let's use LLMs to generate synthetic data" papers to last several conferences. What makes this one interesting - barely - is the political stance detection angle and the somewhat novel SQBC approach. But let's be real here: you're essentially using an LLM to generate slightly different versions of existing viewpoints, then acting surprised when this helps... a little bit.

The authors show that their approach improves F1 scores from ~0.69 to ~0.72 with synthetic data alone, and up to ~0.75 with their full pipeline. Sure, that's positive, but is it worth the computational cost of running Mistral-7B for hours to generate the synthetic data? (And don't get me started on the economic/environmental impact in general - though I suppose that's not this paper's specific sin since those models are small and this used only 1 A100 GPU.)

The most interesting part is actually buried in Section 5.1, where they show that using Mistral-7B directly for stance detection fails miserably. This suggests something important about synthetic data that the authors don't fully explore: it's better at generating plausible variations than at making decisive judgments. This deserved more analysis.

What's missing here is any real investigation into what makes synthetic data actually useful. Are we just doing expensive interpolation between existing data points? Where's the analysis of entropy and diversity in the generated samples? The visualizations in Figure 3 are pretty, but they also show that the synthetic data mostly just fills in obvious gaps rather than introducing genuinely novel perspectives.

The active learning component feels tacked on, though I'll admit the SQBC approach is clever. Using synthetic data as a reference distribution for selecting informative samples is neat, but again - why does this work? The paper handwaves at "ambiguous samples" without diving deeper into the theoretical foundations.

One thing I'll give the authors credit for: they did their homework on the translation pipeline. Using NLLB-330M and actually caring about the quality of the German-English-German round trip is more than many papers bother with. The samples in Table 8 show reasonable quality political discourse generation.

SUGGESTIONS FOR IMPROVEMENT:

- Add analysis of entropy/diversity metrics for synthetic data
- Provide theoretical justification for why synthetic data helps beyond just "more data"
- Compare computational costs vs. benefits more explicitly
- Explore what makes certain synthetic samples more useful than others
- Consider alternative methods for introducing genuine novelty into synthetic data

NITPICKS:

- The abbreviation "SQBC" is used before it's properly defined
- Figure 4 is information-dense to the point of being hard to parse
- Some ablation studies feel perfunctory rather than insightful

CONCLUSION:

This paper is fine. It's not going to revolutionize either synthetic data generation or stance detection, but it makes a modest contribution to both. The experimental work is solid if unexciting, and the results are positive if not earth-shattering. The biggest missed opportunity is not diving deeper into what makes synthetic data actually useful beyond simple interpolation.

The paper should be marginally accepted because it advances the field incrementally and might give others ideas for more innovative approaches. But let's not pretend this is more than a small step forward in a very crowded research space. I don't like these kind of papers

I'd love to see a follow-up that really digs into the entropy question and provides proper theoretical foundations for synthetic data generation in political stance detection. Until then, this feels like another "it works (a bit) but we're not quite sure why" paper.

**Strengths:**

Clean experimental methodology with proper ablation studies
Good visualization of how synthetic data aligns with real data distributions
Actually bothered to translate German political content properly instead of using Google Translate
Reasonable baseline comparisons and honest reporting of limitations
The SQBC approach is somewhat novel, even if not revolutionary

**Weaknesses:**

Limited theoretical justification for why synthetic data helps beyond "moar data good"
Doesn't address the entropy/diversity problem in synthetic data generation
Results are modest (~2-3% improvements) for considerable computational overhead
Heavy reliance on a specific dataset (X-Stance) limits generalizability claims
The "active learning with synthetic data" angle feels like two papers duct-taped together

**Questions:**

Were there any computational/economic reasons for not scaling up your compute? I'm sympathetic to this as I understand the burden of even a single A100 - but if you do have more resources, why not use them?

---

> ### Author Response · Authors · 2024-11-16
>
> Dear reviewer,
>
> We really appreciate your thorough and constructive feedback. We have taken your suggestions to heart and made changes to the best of our abilities. We first want to address your suggestions for improvement and then address general comments about the paper.
>
> > Add analysis of entropy/diversity metrics for synthetic data
>
> > What's missing here is any real investigation into what makes synthetic data actually useful...
>
> * We have now added a more thorough analysis of the synthetic data and what makes it useful:
>     * To measure the diversity, we calculate the entropy of each comment (entropy over words) for both the real and synthetic data, and compare the corresponding distributions.
>     * To gather insights about the diversity in the data, we compare average entropy and average word length for the interquartile ranges of the entropy distribution. The entropy reveals information about the content of the data, while the length acts as surrogate for the structure.
>
> * We have gathered our insights in Section 5.2 and Figure 5:
>    * The results show, that the entropies between the real and synthetic data are similar. We also observe, that the average comment lengths of the synthetic data are much longer than the real data counterparts. Looking at the samples in Appendix C1, it becomes clear that the real comments have a concise writing style, common in online political discussions. The synthetic data comments are verbose, while still addressing the topic in a nuanced way.
>     * Looking back at Figure 3, it becomes apparent that the BERT model seems to be invariant to the comment size, since the synthetic and real data align well despite the difference in comment length. The similar entropies also hints towards a similar amount of diversity in both datasets.
>      * This also aligns with our second experiment which was already in Section 5.2, which we have now renamed as "content vs structure". We showed that fine-tuning with synthetic data that does not align with the topic does work well, which we see as validation for the topic of the synthetic data being important to achieve improvement.
>
> * Thus, one clear insight from this is that at least for the BERT model, we need synthetic data that is aligned with given question(topic).
> * Nonetheless, it would be interesting to explore in future work whether comments that reflect similar structure as the real comments improve performance even further.
>
>
> > Provide theoretical justification for why synthetic data helps beyond just "more data"
>
> * Taking the above into account, we additionally believe the synthetic data helps for a few reasons. We have rewritten the text in the results section to better reflect this:
>   * The synthetic data is generally of high quality: from 10000 generated comments in total, only
>         2%-3% (261) were unusable, while the high quality comments contained nuanced opinions.
>   * We thus can assume and also observe from Figure 3 that the synthetic data effectively interpolates between the real data, smoothening the decision boundary of the BERT model.
>    * This explains why augmenting the synthetic dataset with the most informative samples works so well for increasing synthetic dataset size: the synthetic data reduces overall variance when learning the decision boundary with the most informative samples, helping the model to remain robust while incorporating the new real data.
>
> > Explore what makes certain synthetic samples more useful than others
>
> > Consider alternative methods for introducing genuine novelty into synthetic data
>
> * We believe to have answered this question above, however we also want to restate our thoughts from the introduction. We argue that the synthetic samples are useful on a distributional level:
>   * They serve as prior to the model for the "favor" and "against" classes, since the BERT model is sensitive to content (topic) of the data.
>   * They are useful for detecting samples on the decision boundary of the model such as the most informative samples. This is what inspired SQBC.
>    * While analysing the most informative samples we noticed no clear discernible patterns in the data other
>     than the embeddings being in between the two classes, thus we think a more interesting approach consists in looking at the data from a distributional or statistical learning perspective.
>    * In this sense, we could leverage the structured embedding space provided by the synthetic data to design methods that generate samples outside of this reference distribution. This could be achieved through a latent space approach such as using the synthetic data as latent prior or through an adversarial approach where the generative model attempts to learn  representations that are as far as possible from the reference distribution.
>    * We believe that synthetic data can open up an avenue to designing such methods and have added these ideas to the discussion part (Section 6) of the paper.

---

> ### Author Response · Authors · 2024-11-16
>
> > Compare computational costs vs. benefits more explicitly
>
> * Thank you for this suggestion!
> * We have added a paragraph to the discussion section "Potential impact" stating the advantages and contributions of our method to practitioners in the field of stance detection and online political discussions.
> * In short, good stance detection inference with BERT models is accessible to smaller organisations. The synthetic data generation can be outsourced to specialised infrastructures.
> * The ability to generate useful synthetic data allows to tailor the model to a specific topic, where no data is available and where data collection or labelling efforts are too costly.
>
> > The most interesting part is actually buried in Section 5.1, where they show that using Mistral-7B directly for stance detection fails miserably...
>
> * This is indeed a very interesting result. We have rewritten the section to better detail our thoughts regarding as to why the Mistral-7B model struggles at stance classification with the X-stance dataset:
>   1. The niche topics in the X-Stance dataset are likely not present verbatim in the training data of the Mistral-7B model. While this might also be the case for the SemEval or P-Stance datasets, political tweets about US politicians are more likely to be found online in a similar form.
>   2. The smaller parameter count of the model may affect capturing the overall context when providing it with both a discussion topic and a comment in its context. In contrast, when generating synthetic data, the model only needs to focus on the discussion topic in its context.
>   3. LLMs are usually not fine-tuned to give single word responses. In fact, to get the results in Table 2, we accepted responses that contained the words "favor" or "against". If we specifically matched for "favor" or "against" exclusively, only a handful of comments would fullfil this criteria.
> * Overall, while this result is interesting, it is a bit out of the scope of the paper, since our focus is on shifting stance prediction onto the simpler BERT model.
>
> > Yet another synthetic data paper that shows modest improvements but doesn't quite nail why or how to make synthetic data actually useful. Some interesting ideas buried under conventional methodology.
> * We are glad you found our ideas interesting albeit somewhat buried. Please allow us to restate the motivation behind this paper and why the synthetic data and SQBC approach are connected in the context of online political discussions.
> * We based our work on two main issues in stance detection for online political discussions. 1) Available datasets are usually small or non-existent given how complex and niche online discussion topics can be. 2) In the available datasets there are usually several samples that are hard to classify either because the comments are inherently ambiguous or are simply poorly written. This makes training a good model difficult.
>
>
> > The active learning component feels tacked on, though I'll admit the SQBC approach is clever. Using synthetic data as a reference distribution for selecting informative samples is neat, but again - why does this work? The paper handwaves at "ambiguous samples" without diving deeper into the theoretical foundations
> * Thank you for the suggestion! We hope to now have better highlighted the connection between the approaches in the paper.
> * Regarding the theoretical foundations, we have attempted to provide a bit more theoretical and technical insights in the results and ablations sections and also in the responses above.
> * In case this is not sufficient, we'd be happy to address any additional concerns in this regard.
>
> > Were there any computational/economic reasons for not scaling up your compute? I'm sympathetic to this as I understand the burden of even a single A100 - but if you do have more resources, why not use them?
> * Yes, unfortunately this was the only compute we had available for the project.

---

> > ### Comment · Reviewer_si6z · 2024-11-16
> > **Thank you**
> >
> > Thank you for the explanations and write-up. I may not be a huge fan of these kinds of papers, but you all did the work to be here in my opinion. I have raised my score.

---

### Official Review · Reviewer_cUfQ · 2024-11-03

**Soundness:** 3
**Presentation:** 2
**Contribution:** 2
**Rating:** 6
**Confidence:** 2

**Summary:**

This paper proposes to use LLM-generated synthetic data to augment the training of stance classification models. It also proposes a synthetic data-based active learning method that uses synthetic data to facilitate the selection of unlabelled data for human annotation. Experiments are conducted on the German subset of the X-stance dataset (with the help of machine translation). The results demonstrate that including synthetic data in training can improve stance prediction. The synthetic data-based active learning method, however, is not clearly better than a random selection-based baseline active learning method.

**Strengths:**

- The proposed method is sound. I do not see any major issue with the method.
- Although the idea of using synthetic data to augment models is not entirely new, it probably has not been widely explored for stance prediction.
- The authors conducted extensive experiments to evaluate the method, including varying the size of the synthetic dataset, comparing with meaningful baselines, and the further experiments that compare with a LLM zero-shot baseline.

**Weaknesses:**

- The experiments are conducted using a German dataset, but translation into and back from English is used in order for the method to work (probably because of limited German language understanding and generation capabilities of the Mistral model that is used?) There is no explanation of why the authors do not evaluate the method using an English dataset.
- The novelty and impact of the work is still limited. (1) Using synthetic data to augment models is not new. Although applying the idea to stance prediction might be new, it is one of many NLP tasks. The way synthetic data is generated and used during training in this paper is also standard, hence there is limited technical contribution. (2) The idea of using synthetic data for active learning is very interesting and is novel based on my knowledge. However, its effectiveness is limited based on the experiments. Therefore, overall, although the work is very solid in general, its novelty and impact may not meet the standard of this conference.
- There is room for improvement in terms of presentation. In particular, the active learning method proposed can benefit from first presenting an overview of the high-level intuition behind the method before describing the method itself.

**Questions:**

- It would be very helpful to explain why only a German dataset is used for the experiments. Also, if German text is used, have the authors considered using a different LLM that has good German language processing capabilities for the experiments?

---

> ### Author Response · Authors · 2024-11-16
>
> Dear reviewer,
>
> we would like to thank you for your thorough review and interesting thoughts. In the following, we attempt to answer your questions and comments:
>
> > It would be very helpful to explain why only a German dataset is used for the experiments. Also, if German text is used, have the authors considered using a different LLM that has good German language processing capabilities for the experiments?
> * You are correct! We will add our reasons for using the German X-Stance in the datasets section.
> * In short, we used the German X-Stance dataset because it is the most comprehensive dataset for stance detection in online political discussions. It contains over 48k comments divided into 140 different topics (also called targets). We felt this variety in topics was important to validate our method.
> * Moreover, the individual dataset sizes (per topic) are quite small (between 50-500 samples) which reflects the usual limited data availablility in these kinds of discussion processes.
> * Known English stance detection datasets such as SemEval 2016 or P-Stance have far less varied data. SemEval has only six topics(targets) while P-Stance has three topics(targets). These topics are also not as specific as in the X-stance dataset, since the SemEval and P-Stance datasets mostly contain tweets regarding US politicians and mainstream topics. Therefore, it is more likely the LLM has seen SemEval or P-Stance data in its training set. This is one of the reasons we think the Mistral-7B model performs so poorly on classification on the X-Stance dataset, as the comments there are far less likely to be encountered online.
> * Overall, since we observed during our test that the translation models were able to translate the comments with virtually no loss in quality, we decided to use the German X-Stance dataset.
> * We are also confident that our approaches would work with other languages if the pipeline is adjusted accordingly.
>
> > The novelty and impact of the work is still limited...
>
> * We have added information and insights about the synthetic data in Section 5.2 to further highlight the relevance and importance of our approaches:
>
> * Regarding SQBC, we believe the insight that the synthetic data distribution can act as a reference distribution is novel and can be useful for future work. As stated in the results section, it is particularly interesting that random selection works better than both SQBC and CAL when the synthetic data size is large (M=1000). Our intuition behind this is that with increasing synthetic dataset size, the smoother decision boundary makes the model less sensitive to outliers (which are introduced with the random selection method, but not with SQBC as seen in Figure 4). This robustness induced by the synthetic data allows the model to improve even when selecting random samples. We believe this an interesting insight and result.
>   * It is also important to note that the stance data is relatively homogenous due to its discussion structure as we now show in Section 5.2. This means that if the data were to contain any severe outliers, the random selection method could select these samples and worsen the performance of the model. This would not happen with SQBC due to the k-nearest-neighbors objective.
>
> * We believe that we provide insights for future work with regards to when and how synthetic data is effective in improving stance detection. Apart from Figure 3 and the discussion around it, we have  added further technical and theoretical insights in the results section. We have also added a new ablation where we inspect the entropy distribution of the data together with the corresponding average lengths of the comments in Section 5.2 (see Figure 5). We sum the insights up briefly:
>   * The synthetic data is generally of high quality: from 10000 generated comments in total, only 2%-3% (261) were unusable, while the high quality comments contained nuanced opinions.
>    * We observe that despite the synthetic comments being longer than the real comments, the entropy is very similar. Also as seen in Figure 3, the synthetic and real data align well, which indicates that the BERT model is invariant to comment length which can be seen as a surrogate for comment structure.
>    * Together with the "Content vs Structure" ablation, this validates the observation that a BERT based stance detection model is more sensitive to the actual content of the data than the structure.
>
> * We have also added a section regarding the potential impact of our work for the area of stance detection in real world settings:
>   * In short, good stance detection inference with BERT models is accessible to smaller organisations. The synthetic data generation can be outsourced to specialised infrastructures.
>   * The ability to generate useful synthetic data allows to tailor the model to a specific topic, where no data is available and where data collection or labelling efforts are too costly.

---

> > ### Comment · Reviewer_cUfQ · 2024-11-17
> > **Thank you for addressing some of my questions!**
> >
> > It's nice to see the elaboration on why the German dataset was used instead of an English dataset. I still have some reservation regarding the novelty and impact of the work (as another reviewer also commented on), but I'm happy to slightly raise my score.

---

> ### Author Response · Authors · 2024-11-16
>
> >  There is room for improvement in terms of presentation. In particular, the active learning method proposed can benefit from first presenting an overview of the high-level intuition behind the method before describing the method itself.
> * Thank you for this suggestion!
> * We have added to the beginning of the method section to better connect the concrete method to the high level intuition from the introduction section and to Figure 1.

---

### Official Review · Reviewer_kmPK · 2024-11-03

**Soundness:** 3
**Presentation:** 2
**Contribution:** 3
**Rating:** 8
**Confidence:** 3

**Summary:**

The paper tries to improve transformer-based stance detection models by fine-tuning on LLM generated data. They compare the real-world data with the synthetic data to identify difficult samples from unlabelled data (active learning) to further improve the model. They show that both these steps improves the performance of the transformer-based baseline.

**Strengths:**

- The paper leverages the data augmentation capabilities of LLMs to improve transformer based models which are better suited for online deployment as they are more reliable.
- The presented method can be adapted to other text classification tasks and hence is a significant contribution.
- It is well written and easy to follow, except for few instances mentioned in the comments.

**Weaknesses:**

- It’s possible I’m missing some key context here, but I’m having trouble following the ablation study in Section 5.2. To test whether the performance gains come from dataset size or the generated content itself, the authors “shuffle” instances, apparently misaligning the posed questions with synthetic data. If the synthetic data consists of single text instances with labels, this shuffling wouldn’t seem to affect outcomes. Perhaps the authors mean they’re using different proportions of synthetic data in each run while keeping the total instance count constant, but this explanation feels somewhat unclear.
- Even though authors acknowledge this as a limitation, fine-tuning a separate model for each question doesnot seem to be a scalable approach, especially when the main motivation for the research was in line with training robust models for online deployment.
- The X-stance dataset is described as having around 48k annotated comments on various questions. However, an overview of the dataset’s statistics—such as the number of comments per question—would greatly enhance readability. When you mention selecting 10 questions from the test set, it would be helpful to specify how many comments correspond to each question. While I see some statistics are included in the Appendix, a high-level summary within the main text would improve clarity and context for readers.
- Section 4.2, General setup: Please review this section for more readability. Currently, it is a bit difficult to get a picture of what models are being tested and how the methods differ between them.

**Questions:**

- Why choose translation over adapting prompts directly? Is Mistral unable to generate responses in German, or were other multilingual models considered?

- In the ablation study for "Content vs. Size", I am not sure I understand why you call the shuffled dataset "misaligned". Could you please explain the reasoning behind this?

- In the generated dataset, did you find any instance where the LLM failed to generate the requested content? For instance, generate statements not in favor when requested for "in favor" content or LLM refusing to generate any relevant content at all.

---

> ### Author Response · Authors · 2024-11-16
>
> Dear reviewer,
>
> we thank you for putting in the time to provide us with a thorough review. We are eager to respond to your questions and comments:
>
> > It’s possible I’m missing some key context here, but I’m having trouble following the ablation study in Section 5.2..
>
> > In the ablation study for "Content vs. Size", I am not sure I understand why you call the shuffled dataset "misaligned". Could you please explain the reasoning behind this.
>
> * Thank you for pointing this out!
> * We do not shuffle instances within the synthetic dataset. We shuffle the entire datasets with regard to the questions, i.e., we shuffle the correspondences between question and (Q) and synthetic dataset (SD).
> * For instance, we have 10 Questions and therefore 10 corresponding synthetic datasets. Instead of having the natural correspondence of Q1->SD1, Q2->SD2 ….. and so on, we change the correspondences, e.g., Q1->SD9, Q2->SD5, ….
> * Overall, we have renamed the section "content vs structure" and attempted to describe the purpose of this ablation more clearly. The core idea is to analyse whether per-topic fine-tuning is really necessary. For this, as described above, we change the correspondences of the given question and the synthetic datasets for fine-tuning.
> * We have also added a new ablation in the same Section which shows the entropy distribution of the data and the average length of comments. We observe that despite the synthetic comments being longer than the real comments, the entropy is very similar. Also as seen in Figure 3, the synthetic and real data align well, which indicates that the BERT model
> is invariant to comment length which can be seen as a surrogate for comment structure.
> * Together with the "Content vs Structure" ablation, this further validates the observation that a BERT based stance detection model is more sensitive to the actual content of the data than the structure.
>
> >  Even though authors acknowledge this as a limitation, fine-tuning a separate model for each question does not seem to be a scalable approach, especially when the main motivation for the research was in line with training robust models for online deployment.
> * We understand this goes against the current paradigm of training large general models. However, there are a number of benefits to fine-tuning separate models in an online political discussion context:
>   * They are more efficient to deploy for inference, especially for smaller organisations that do not have many resources.
>   * Considering participation processes in government, these are usually carried out on a per topic basis.
>   * Concerning robustness, together with synthetic data we can tailor models to specific topics while outsourcing the synthetic data generation to larger infrastructure.
>   * The smaller stance detection model remains interpretable and more predictable compared to larger general models as we show in Section 5.1.
> * The new ablation in Section 5.2 further highlights the importance of topic alignment between the model and the synthetic data. We added these points for further clarity to the discussion section.
>
> > The X-stance dataset is described as having around 48k annotated comments on various questions. However, an overview of the dataset’s statistics—such as the number of comments per question—would greatly enhance readability. When you mention selecting 10 questions from the test set, it would be helpful to specify how many comments correspond to each question. While I see some statistics are included in the Appendix, a high-level summary within the main text would improve clarity and context for readers.
> * We have added more information regarding the amount of topics and a table in the datasets section of the exact sizes and splits of our test datasets.
>
> > Section 4.2, General setup: Please review this section for more readability. Currently, it is a bit difficult to get a picture of what models are being tested and how the methods differ between them.
> * Again, thank you for pointing this out! We have improved this section accordingly.
>
>
> >  Why choose translation over adapting prompts directly? Is Mistral unable to generate responses in German, or were other multilingual models considered?
> * Yes, we found that smaller models such as Mistral-7B struggle at generating grammatically correct and coherent sentences in German. At the time of performing our experiments, among the smaller models we did not manage to find a good German model to generate high quality comments.
> * In any case, we found that there is virtually no loss in quality when using the translation model.

---

> ### Author Response · Authors · 2024-11-16
>
> >  In the generated dataset, did you find any instance where the LLM failed to generate the requested content? For instance, generate statements not in favor when requested for "in favor" content or LLM refusing to generate any relevant content at all.
> * We have added more insights regarding the synthetic data in Figure 5(Right) and Appendix E1.
> * Only 2-3% of all synthetic comments were unusable, in these samples the LLM would either generate a very short comment or repeat certain content for a number of times until it would catch itself to finish writing the comment. Appendix C1 shows some of these samples withe minimum and maximum entropy.
> * We also did not find instances in the synthetic data where a comment with the opposing stance was generated. We believe the request of generating a comment in "favor" or "against" is a strong conditioning variable for the model.
> * Of course, with different models mileage may vary. With the rapid advancements and the plethora of models available, we chose not to focus our work on an ablation of the different models. Instead we wanted to showcase a small model that works well and is relatively efficient, while feasible to run.

---

> > ### Comment · Reviewer_kmPK · 2024-11-17
> > **Thank you for the detailed response**
> >
> > It is nice to see that all the concerns raised by the reviewers are addressed. I have raised my score based on the response.

---

### Author Response · Authors · 2024-11-16
**Revision**

Dear reviewers,

we want to thank you for your thorough and detailed reviews. We believe your suggestions have improved our paper. In the following, we present the changes made to the revision which are marked in Blue. We have also responded to each of your concerns under your respective reviews.

* We have made some slight rewrites in the introduction to strengthen the connection between fine-tuning and SQBC.
We have restructured the experimental setup page, added information about the X-Stance dataset(and why we use it), and included a new table with the exact test data splits.
* We have rewritten the (Q2) and (Q3) in the results section (Section 4.3) to better reflect our insights regarding as to what synthetic data does to the model and regarding the behaviour of the active learning methods. We hope this addresses the request for more technical and theoretical insights.
* We have added a new Figure 5 and a new paragraph to Section 5.2. We now analyse the synthetic data by plotting the entropy distribution of the synthetic and the real data, while also calculating the average entropy and comment length for the interquartile ranges of the entropy distribution. Furthermore, we also show comment samples for both synthetic and real data in Appendix C1 for the interquartile ranges of the entropy distributions. The idea behind this is to show together with Figure 3 that the BERT model is largely invariant to comment structure and is more sensitive to the content (i.e. topic) of the data.
* We therefore renamed the "content vs size" paragraph to "content vs structure: Is per topic fine-tuning necessary?". We did this to better highlight the idea behind this ablation, which is to show together with the synthetic data analysis that the effectiveness of fine-tuning with synthetic data is largely due to the content of the comments.
* Finally, we have added two paragraphs to the discussion section. First, we added "Potential impact.", where we highlight the benefits of using our analysis for practitioners in the are of stance detection in online political discussions. Second, we added a section about future work to better highlight open problems arising from our work.

---

### Author Response · Authors · 2024-11-25

We want to thank all reviewers again for their feedback and are glad that the changes made to the paper were received positively!

We have uploaded another revision, improving minor nitpicks such as improving figure and table readability (improving captions) and correcting typos.

---

### Meta-Review · Area_Chair_nP8R · 2024-12-19

**Metareview:**

**Summary:**

The authors explore the efficacy of using LLM-generated synthetic training data for automated stance detection. They primarily test against a Bert baseline where individual models are trained for different stance detection question types. Varying levels of synthetic data are used to finetune this model, as well as previously unlabeled human data collected using active learning that makes use of nearest neighbor synthetic examples. Results indicate performance gains from synthetic data (72.3 vs 69.3 F1) and synthetic data mixed with active learning labeled data (75.4 F1).

**Strengths:**

- The authors' research questions are well-defined and the results seem to support use of synthetic data for stance detection

- While the findings are unsurprising given recent literature around effectiveness of synthetic data on other tasks like content moderation, the experimentation is very solid and there are unique aspects to their approach such as using synthetic nearest-neighbors to facilitate active learning on human-derived data.

**Weaknesses:**

- The experimental results are not very comprehensive and only consider the effect on a Bert baseline. The authors also test against Mistral, but only in a zero-shot vs fine-tuned setting. Given the sensitivity of LLMs to prompt phrasing, much more could be explored here including few-shot demonstrations in the prompt or specifying the desired output format.

- The authors don't seem to conduct any analysis to ensure the synthetic comments are consistent with the data labels used in prompting.

- The paper would benefit from statistically significance testing, in particular when comparing baseline vs. synthetic only results.

I think the paper is technically sound enough to be accepted and commend the authors on their efforts, however I question the impact the paper's appearance in conference proceedings will make on the broader research community.

**Additional Comments On Reviewer Discussion:**

The reviewers unanimously agree that the paper should be accepted. Primary concerns were (1) clarity issues with the presentation of results, particularly ablations, that have been resolved in the revised paper, and (2) lack of analysis on why synthetic data works, which the authors addressed by expanding the discussion to cover entropy/diversity. There are still lingering concerns, raised by reviewer cUfQ, that the technical contribution is minimal and not befitting of ICLR.

---

### Decision · Program_Chairs · 2025-01-22

Accept (Spotlight)